# Multi-task Learning of Order-Consistent Causal Graphs

**Xinshi Chen**[*]
Georgia Institute of Technology
xinshi.chen@gatech.edu

**Haoran Sun**
Georgia Institute of Technology
haoransun@gatech.edu

**Caleb Ellington**
Carnegie Mellon University
cellingt@cs.cmu.edu

**Eric Xing**
Carnegie Mellon University
MBZUAI
eric.xing@mbzuai.ac.ae

**Le Song**
BioMap
MBZUAI
le.song@mbzuai.ac.ae

## Abstract

We consider the problem of discovering $K$ related Gaussian directed acyclic graphs (DAGs), where the involved graph structures share a consistent causal order and sparse unions of supports. Under the multi-task learning setting, we propose a $l_1/l_2$-regularized maximum likelihood estimator (MLE) for learning $K$ linear structural equation models. We theoretically show that the joint estimator, by leveraging data across related tasks, can achieve a better sample complexity for recovering the causal order (or topological order) than separate estimations. Moreover, the joint estimator is able to recover non-identifiable DAGs, by estimating them together with some identifiable DAGs. Lastly, our analysis also shows the consistency of union support recovery of the structures. To allow practical implementation, we design a continuous optimization problem whose optimizer is the same as the joint estimator and can be approximated efficiently by an iterative algorithm. We validate the theoretical analysis and the effectiveness of the joint estimator in experiments.

## 1 Introduction

Estimating causal effects among a set of random variables is of fundamental importance in many disciplines such as genomics, epidemiology, health care and finance [1, 2, 3, 4, 5, 6]. Therefore, designing and understanding methods for causal discovery is of great interests in machine learning.

Causal discovery from finite observable data is often formulated as a directed acyclic graph (DAG) estimation problem in graphical models. A major class of DAG estimation methods are score-based, which search over the space of all DAGs for the best scoring one. However, DAG estimation remains a very challenging problem from both the *computational* and *statistical* aspects [7]. On the one hand, the number of possible DAG structures grows super-exponentially in the number of random variables, whereas the number of observational sample size is normally small. On the other hand, some DAGs are *non-identifiable* from observational data even with infinitely many samples.

Fortunately, very often multiple related DAG structures need to be estimated from data, which allows us to leverage their similarity to improve the estimator. For instance, in bioinformatics, gene expression levels are often measured over patients with different subtypes [8, 9] or under various experimental conditions [10]. In neuroinformatics, fMRI signals are often recorded for multiple subjects for studying the brain connectivity network [11, 12]. In these scenarios, multiple datasets will be collected, and their associated DAGs are likely to share similar characteristics. Intuitively, it may be beneficial to estimate these DAGs jointly.

---

[*]Work done partially during the visit at MBZUAI (Mohamed bin Zayed University of Artificial Intelligence)

35th Conference on Neural Information Processing Systems (NeurIPS 2021).

In this paper, we focus on the analysis of the multi-task DAG estimation problem where the DAG structures can be related by a consistent causal order and a (partially) shared sparsity pattern, but allowed to have different connection strengths and edges, and differently distributed variables. In this setting, we propose a joint estimator for recovering multiple DAGs based on a group norm regularization. We prove that the joint $l_1/l_2$-penalized maximum likelihood estimator (MLE) can recover the causal order better than individual estimators.

Intuitively, it is not surprising that joint estimation is beneficial. However, our results provide a quantitative characterization on the improvement in sample complexity and the conditions under which such improvement can hold. We show that:

- For identifiable DAGs, if the shared sparsity pattern (union support) size $s$ is of order $\mathcal{O}(1)$ in $K$ where $K$ is the number of tasks (DAGs), then the *effective* sample size for order recovery will be $nK$ where $n$ is the sample size in each problem. Furthermore, as long as $s$ is of order $o(\sqrt{K})$ in $K$, the joint estimator with group norm regularization leads to an improvement in sample complexity.
- A non-identifiable DAG cannot be distinguished by single-task estimators even with indefinitely many observational data. However, non-identifiable DAGs can be recovered by our joint estimator if they are estimated together with other identifiable DAGs.

Apart from the theoretical guarantee, we design an efficient algorithm for approximating the joint estimator through a formulation of the combinatorial search problem to a continuous programming. This continuous formulation contains a novel design of a learnable masking matrix, which plays an important role in ensuring the acyclicity and shared order for estimations in all tasks. An interesting aspect of our design is that we can learn the masking matrix by differentiable search over a continuous space, but the optimum must be contained in a discrete space of cardinality $p!$ (reads $p$ factorial, where $p$ is the number of random variables).

We conduct a set of synthetic experiments to demonstrates the effectiveness of the algorithm and validates the theoretical results. Furthermore, we apply our algorithm to more realistic single-cell expression RNA sequencing data generated by SERGIO [13] based on real gene regulatory networks.

The remainder of the paper is organized as follows. In Section 2, we introduce the linear structural equation model (SEM) interpretation of Gaussian DAGs and its properties. Section 3 is devoted to the statement of our main results, with some discussion on their consequences and implications. In Section 4, we present the efficient algorithm for approximating the joint estimator. Section 5 summarizes related theoretical and practical works. Experimental validations are provided in Section 6.

## 2 Background

A substantial body of work has focused on the linear SEM interpretation of Gaussian DAGs [14, 15, 16, 17]. Let $X = (X_1, \cdots, X_p)$ be a $p$-dimensional random variable. Then a linear SEM reads

$$X = \widetilde{G}_0^\top X + W, \quad W \sim \mathcal{N}(0, \Omega_0), \tag{1}$$

where $\Omega_0$ is a $p \times p$ positive diagonal matrix which indicates the variances of the noise $W$. $\widetilde{G}_0 \in \mathbb{R}^{p \times p}$ is the connection strength matrix or adjacency matrix. Each nonzero entry $\widetilde{G}_{0ij}$ represents the direct causal effect of $X_i$ on $X_j$. This model implies that $X$ is Gaussian, $X \sim \mathcal{N}(0, \Sigma)$, where

$$\Sigma := (I - \widetilde{G}_0)^{-\top} \Omega_0 (I - \widetilde{G}_0)^{-1}. \tag{2}$$

**Causal order $\pi_0$.** The nonzero entries of $\widetilde{G}_0$ defines its causal order (also called topological order), which informs possible "parents" of each variable. A causal order can be represented by a permutation $\pi_0$ over $[p] := (1, 2, \cdots, p)$. We say $\widetilde{G}_0$ is **consistent** with $\pi_0$ if and only if

$$\widetilde{G}_{0ij} \neq 0 \Rightarrow \pi_0(i) < \pi_0(j). \tag{3}$$

There could exist more than one permutations that are consistent with a DAG structure $\widetilde{G}_0$, so we denote the set of permutations that satisfy Eq. 3 by $\Pi_0$. Once a causal order $\pi_0$ is identified, the connection strengths $\widetilde{G}_0$ can be estimated by ordinary least squares regression which is a comparatively easier problem.

**Identifiability and equivalent class.** However, estimating the true causal order $\pi_0$ is very challenging, largely due to the existence of the equivalent class described below.

Let $\mathbb{S}_p$ be the set of all permutations over $[p]$. For every $\pi \in \mathbb{S}_p$, there exists a connection strength matrix $\widetilde{G}(\pi)$, which is consistent with $\pi$, and a diagonal matrix $\Omega(\pi) = \mathrm{diag}\left[\sigma_1(\pi)^2, \cdots, \sigma_p(\pi)^2\right]$ such that the variance $\Sigma$ in Eq. 2 equals to

$$\Sigma = (I - \widetilde{G}(\pi))^{-\top}\Omega(\pi)(I - \widetilde{G}(\pi))^{-1}, \tag{4}$$

and therefore the random variable $X$ in Eq. 1 can be equivalently described by the model [15]:

$$X = \widetilde{G}(\pi)^\top X + W(\pi), \quad W(\pi) \sim \mathcal{N}(0, \Omega(\pi)). \tag{5}$$

Without further assumption, the true underlying DAG, $\widetilde{G}_0 = \widetilde{G}(\pi_0)$, **cannot** be identified from the

$$\text{equivalent class:} \quad \{\widetilde{G}(\pi) : \pi \in \mathbb{S}_p\} \tag{6}$$

based on the distribution of $X$, even if infinitely many samples are observed.

## 3 Joint estimation of multiple DAGs

In the multi-task setting, we consider $K$ linear SEMs:

$$X^{(k)} = \widetilde{G}_0^{(k)\top} X^{(k)} + W^{(k)}, \quad \text{for } k = 1, \cdots, K.$$

The superscript notation $^{(k)}$ indicates the $k$-th model. As mentioned in Sec 2, each model defines a random variable $X^{(k)} \sim \mathcal{N}(0, \Sigma^{(k)})$ with $\Sigma^{(k)} = (1 - \widetilde{G}_0^{(k)})^{-\top}\Omega_0^{(k)}(1 - \widetilde{G}_0^{(k)})^{-1}$.

### 3.1 Assumption

**Similarity.** In Condition 3.1 and Definition 3.1, we make a few assumptions about the similarity of these models. Condition 3.1 ensures the causal orders in the $K$ DAGs do not have conflicts. Definition 3.1 measures the similarity of the sparsity patterns in the $K$ DAGs. If the sparsity patterns are strongly overlapped, then the size of the support union $s$ will be small. We do not enforce a strict constraint on the support union, but we will see in the later theorem that a smaller $s$ can lead to a better recovery performance.

*Condition* 3.1 (Consistent causal orders). There exists a nonempty set of permutations $\Pi_0 \subseteq \mathbb{S}_p$ such that $\forall \pi_0 \in \Pi_0$, it holds for all $k \in [K]$ that $\widetilde{G}_{0ij}^{(k)} \neq 0 \Rightarrow \pi_0(i) < \pi_0(j)$.

*Definition* 3.1 (Support union). Recall $\widetilde{G}(\pi)$ in Eq. 6. The support union of the $K$ DAGs associated with permutation $\pi$ is denoted by $S(\pi) := \left\{(i,j) : \exists k \in [K] \, s.t. \, \widetilde{G}_{ij}^{(k)}(\pi) \neq 0\right\}$. The support union of the $K$ true DAGs $\widetilde{G}_0^{(k)}$ is $S_0 := S(\pi_0)$. We further denote $s_0 := |S_0|$ and $s := \sup_{\pi \in \mathbb{S}_p} |S(\pi)|$.

**Identifiability.** To ensure the consistency of the estimator based on least squared loss, we first assume that the DAGs to be recovered are minimum-trace DAGs.

*Condition* 3.2 (Minimum-trace). Recall the equivalent class defined in Eq. 4 to Eq. 4. Assume for all $k = 1, \cdots, K$, $\mathrm{trace}(\Omega_0^{(k)}) = \min_{\pi \in \mathbb{S}_p} \mathrm{trace}(\Omega(\pi)^{(k)})$.

However, the minimum-trace DAG may not be unique without further assumptions, making the true DAG indistinguishable from other minimum-trace DAGs. Therefore, we consider the equal variance condition in Condition 3.3 which ensures the uniqueness of the minimum-trace DAG. In this paper, we assume the first $K' \leq K$ models satisfy this condition, so they are identifiable. We do not make such an assumption on the other $K - K'$ models, so the $K - K'$ models may not be identifiable.

*Condition* 3.3 (Equal variance). For all $k = 1, \cdots, K'$ with $K' \leq K$, the noise $W^{(k)} \sim \mathcal{N}(0, \Omega_0^{(k)})$ has equal variance with $\Omega_0^{(k)} = \sigma_0^{(k)} I_p$.

### 3.2 $l_1/l_2$-penalized joint estimator

Denote the sample matrix by $\boldsymbol{X}^{(k)}$ whose row vectors are $n$ i.i.d. samples from $\mathcal{N}(0, \Sigma^{(k)})$. Based on the task similarity assumptions, we propose the following $l_1/l_2$-penalized joint maximum likelihood estimator (MLE) for jointly estimating the connection strength matrices $\{\widetilde{G}_0^{(k)}\}_{k=1}^K$:

$$\hat{\pi}, \{\widehat{G}^{(k)}\} = \underset{\pi \in \mathbb{S}_p, \{G^{(k)} \in \mathbb{D}(\pi)\}}{\arg\min} \sum_{k=1}^K \frac{1}{2n}\|\boldsymbol{X}^{(k)} - \boldsymbol{X}^{(k)}G^{(k)}(\pi)\|_F^2 + \lambda\|G^{(1:K)}(\pi)\|_{l_1/l_2}. \tag{7}$$

Similar to the notation in Eq. 4, $G^{(k)}(\pi)$ indicates its consistency with $\pi$. $\mathbb{D}(\pi)$ denote the space of all DAGs that are consistent with $\pi$. It is notable that a single $\pi$ shared across all $K$ tasks is optimized in Eq. 7, which respects Condition 3.1. The group norm over the set of $K$ matrices is defined as

$$\|G^{(1:K)}(\pi)\|_{l_1/l_2} := \sum_{i=1}^{p} \sum_{j=1}^{p} \|G_{ij}^{(1:K)}\|_2, \quad \text{where } G_{ij}^{(1:K)} := [G_{ij}^{(1)}(\pi), \cdots, G_{ij}^{(K)}(\pi)].$$

It will penalize the size of union support in a soft way. When $K = 1$, this joint estimator will be reduced to the $l_1$-penalized maximum likelihood estimation.

*Remark* 3.1. The optimization in Eq. 7 is used for analysis only. A continuous program with the same optimizer will be discussed in Section 4 for practical implementation.

## 3.3 Main result: causal order recovery

We start with a few pieces of notations and definitions. Then the theorem statement follows.

*Definition* 3.2. Let $\widetilde{g}_j^{(k)}(\pi)$ denote the $j$-th column of $\widetilde{G}^{(k)}(\pi)$. Let

$$d := \sup_{j\in[p],\pi\in\mathbb{S}_p} \Big| \cup_{k\in[K]} \mathrm{supp}(\widetilde{g}_j^{(k)}(\pi)) \Big|, \quad g_{\max} := \sup_{\pi\in\mathbb{S}_p,(i,j)\in S(\pi)} \|\widetilde{G}_{ij}^{(1:K)}(\pi)\|_2/\sqrt{K}.$$

In Definition 3.2, $d$ is the maximal number of parents (in union) in the DAGs $\widetilde{G}^{(k)}(\pi)$, which is also a measure of the sparsity level. $g_{\max}$ is bounded by the maximal entry value in the matrices $\widetilde{G}^{(k)}(\pi)$.

*Condition* 3.4 (Bounded spectrum). Assume for all $k = 1, \cdots, K$, the covariance matrix $\Sigma^{(k)}$ is positive definite. There exists constants $0 < \Lambda_{\min} \le \Lambda_{\max} < \infty$ such that for all $k = 1, \cdots, K$,

(a) all eigenvalues of $\Sigma^{(k)}$ are upper bounded by $\Lambda_{\max}$;
(b) all eigenvalues of $\Sigma^{(k)}$ are lower bounded by $\Lambda_{\min}$.

*Condition* 3.5 (Omega-min). There exists a constant $\eta_w > 0$ such that for any permutations $\pi \notin \Pi_0$,

$$\frac{1}{pK'} \sum_{k=1}^{K'} \sum_{j=1}^{p} (\sigma_j^{(k)}(\pi)^2 - \sigma_0^{(k)2})^2 > \frac{1}{\eta_w}.$$

Condition 3.5 with $K' = K = 1$ is called 'omega-min' condition in [14], so we follows this terminology. In some sense, when $\eta_w$ is larger, $\sigma_j(\pi)$ with $\pi \notin \Pi_0$ is allowed to deviate less from the true variance $\sigma_j(\pi_0) = \sigma_0$ with $\pi_0 \in \Pi_0$, which will make it more difficult to separate the set $\Pi_0$ from its complement in a finite sample scenario. Ideally, we should allow $\eta_w$ to be large, so that the recovery is not only restricted to easy problems.

Now we are ready to present the recovery guarantee for causal order. Theorem 3.1 is a specific statement when the regularization parameter $\lambda$ follows the classic choice in (group) Lasso problems. A more general statement which allows other $\lambda$ is given in Appendix B along with the proof.

**Theorem 3.1** (Causal order recovery). *Suppose we solve the joint optimization in Eq. 7 with specified regularization parameter $\lambda = \sqrt{\frac{p\log p}{n}}$ for a set of $K$ problems that satisfy Condition 3.1, 3.4 (a), 3.2, 3.3 and 3.5. If the following conditions are satisfied:*

$$\theta(n, K, K', p, s) := \frac{p}{s}\sqrt{\frac{n}{p\log p}\frac{K'^2}{K}} > \kappa_1\eta_w, \tag{8}$$

$$n \ge c_1 \log K + c_2(d+1)\log p, \tag{9}$$

$$K \le \kappa_2 p \log p, \tag{10}$$

*then the following statements hold true:*

(a) *With probability at least $1 - c_3\exp(-\kappa_4(d+1)\log p) - \exp(-c_4 n)$, it holds that*

$$\hat{\pi} \in \Pi_0.$$

(b) *If in addition, $n$ satisfies $n \ge \kappa_5\hat{d}(\log K + \log p)$ with $\hat{d} := \max_{j\in[p],k\in[K]}\|\hat{g}_j^{(k)} - \widetilde{g}_j^{(k)}\|_0$, and Condition 3.4 (b) holds, then with probability at least $1 - c_3\exp(-\kappa_4(d+1)\log p) - \exp(-c_4 n) - \exp(-\log p - \log K)$,*

$$\frac{1}{K}\sum_{k=1}^{K}\|\widehat{G}^{(k)} - \widetilde{G}_0^{(k)}\|_F^2 = \mathcal{O}\left(s_0\sqrt{\frac{p\log p}{nK}}\right).$$

In this statement, $c_1, c_2, c_3, c_4$ are universal constants (i.e., independent of $n, p, K, s, \Sigma^{(k)}$), $\kappa_1, \kappa_2, \kappa_3, \kappa_4$ are constants depending on $g_{\max}$ and $\Lambda_{\max}$, and $\kappa_5$ is a constant depending on $\Lambda_{\min}$.

**Discussion on Eq. 8-Eq. 10.** (i) In Eq. 8, Theorem 3.1 identifies a *sample complexity parameter* $\theta(n, K, K', p, s)$. Following the terminology in [18], our use of the term "sample complexity" for $\theta$ reflects the dominant role it plays in our analysis as the rate at which the sample size much grow in order to obtain a consistent causal order. More precisely, for scalings $(n, K, K', p, s)$ such that $\theta(n, K, K', p, s)$ exceeds a fixed critical threshold $\kappa_1 \eta_w$, we show that the causal order can be correctly recovered with high probability.

(ii) The additional condition for the sample size $n$ in Eq. 9 is the sample requirement for an ordinary linear regression problem. It is in general much weaker than Eq. 8, unless $K$ grows very large.

(iii) The last condition in Eq. 10 on $K$ could be relaxed if a tighter analysis on the distribution properties of Chi-squared distribution is available. However, it is notable that this restriction on the size of $K$ has already been weaker than many related works on multi-task $\ell_1$ sparse recovery, which either implicitly treat $K$ as a constant [18, 19] or assume $K = o(\log p)$ [20, 21].

**Recovering non-identifiable DAGs.** A direct consequence of Theorem 3.1 is that as long as the number of identifiable DAGs $K'$ is non-zero, the joint estimator can recover the causal order of non-identifiable DAGs with high probability. This is not achievable in separate estimation even with infinitely many samples. Therefore, we show that how the information of identifiable DAGs helps recover the non-identifiable ones.

**Effective sample size.** As indicated by $\theta(n, K, K', p, s)$ in Eq. 8, the effective sample size for recovering the correct causal order is $\frac{nK'^2}{K}$ if the support union size $s$ is of order $\mathcal{O}(1)$ in $K$. To show the improvement in sample complexity, it is more fair to consider the scenario when the DAGs are identifiable, i.e., $K' = K$. In this case, it is clear that the parameter $\theta(n, K, K, p, s)$ indicates a lower sample complexity relative to separate estimation as long as $s$ is of order $o(\sqrt{K})$.

**Separate estimation.** Consider the special case of a single task estimation with $K = K' = 1$, in which the joint estimator reduces to $\ell_1$-penalized MLE. We discuss how our result can recover previously known results for single DAG recovery. Unfortunately, existing analyses were conducted under different frameworks with different conditions. [14] and [22] are the most comparable ones since they were based on the same omega-min condition (i.e., Condition 3.5), but they chose a smaller regularization parameter $\lambda$. In our proof, the sample complexity parameter is derived from $\theta(n, K, K', p, s) = pK' / \left( s\lambda\sqrt{K} \right)$ and Eq. 8 is $pK' / \left( s\lambda\sqrt{K} \right) > \kappa_1 \eta_w$. When $K = K' = 1$, this condition matches what is identified in [14] and [22] for recovering the order of a single DAG.

**Error of $\widehat{G}^{(k)}$.** To compare the estimation of $\widehat{G}^{(k)}$ to the true DAG $\widetilde{G}_0^{(k)}$, Theorem 3.1 (b) says the averaged error in F-norm goes to zero when $nK \to \infty$. It decreases in $K$ as long as $s = o(K)$.

To summarize, Theorem 3.1 analyzes the causal order consistency for the joint estimator in Eq. 7. Order recovery is the most challenging component in DAG estimation. After $\hat{\pi} \in \Pi_0$ has been identified, the DAG estimation becomes $p$ linear regression problems that can be solved separately. Theorem 3.1 (b) only shows the estimation error of connection matrices in F-norm. To characterize the structure error, additional conditions are required.

### 3.4 Support union recovery

Theorem 3.1 has shown that $\hat{\pi} = \pi_0 \in \Pi_0$ holds with high probability. Consequently, the support recovery analysis in this section is conditioned on this event. In fact, given the true order $\pi_0$, what remains is a set of $p$ separate $l_1/l_2$-penalized group Lasso problems, in each of which the order $\pi_0$ plays a role of constraining the support set by the set $S_j(\pi_0) := \{i : \pi_0(i) < j\}$. However, we need to solve $p$ such problems simultaneously where $p$ is large. A careful analysis is required, and directly combining existing results will not give a high recovery probability.

In the following, we impose a set of conditions and definitions, which are standard in many $l_1$ sparse recovery analyses [18, 19], after which theorem statement follows. See Appendix C for the proof.

*Definition* 3.3. The union support of $j$-th columns of $\{\widetilde{G}_0^{(k)}\}_{k \in [K]}$ is denoted by $\mathrm{RS}_j := \{i \in [p] : \exists k \in [K] \, s.t. \, \widetilde{G}_{0ij}^{(k)} \neq 0\}$. The maximal cardinality is $r_{\max} := \sup_{j \in [p]} \left| \mathrm{RS}_j \right|$.

*Definition* 3.4. $\rho_u := \sup_{k\in[K], j\in[p], S=\mathrm{RS}_j} \max_{i\in S^c} \left(\Sigma_{S^c S^c|S}^{(k)}\right)_{ii}$ is the maximal diagonal entry of the conditional covariance matrices, where $\Sigma_{S^c S^c|S}^{(k)} := \Sigma_{S^c S^c}^{(k)} - \Sigma_{S^c S}^{(k)}(\Sigma_{SS}^{(k)})^{-1}\Sigma_{SS^c}^{(k)}$.

*Definition* 3.5. $g_{\min} := \inf_{(i,j)\in S_0} \|\widetilde{G}_{0ij}^{(1:K)}\|_2/\sqrt{K}$ represents the signal strength.

*Condition* 3.6 (Irrepresentable condition). There exists a fixed parameter $\gamma \in (0,1]$ such that $\sup_{j\in[p], S=\mathrm{RS}(\widetilde{g}_{0j}^{(1:K)})} \|A(S)\|_\infty \le 1-\gamma$, where $A(S)_{ij} := \sup_{k\in[K]} \left|\left(\Sigma_{S^c S}^{(k)}(\Sigma_{SS}^{(k)})^{-1}\right)_{ij}\right|$.

**Theorem 3.2** (Union support recovery). *Assume on the subset of probability space where $\{\hat{\pi} \in \Pi_0\}$ holds, and assume Condition 3.6. Assume the following conditions are satisfied*

$$n \ge \kappa_6 r_{\max} \log p, \tag{11}$$

$$K \le c_7 \log p, \tag{12}$$

$$\sqrt{\frac{8\Lambda_{\max} \log p}{\Lambda_{\min} n}} + \frac{2}{\Lambda_{\min}}\sqrt{\frac{p \log p}{n}\frac{r_{\max}}{K}} = o(g_{\min}), \tag{13}$$

*where $\kappa_6$ is a constant depending on $\gamma, \Lambda_{\min}, \rho_u, \sigma_{\max}$, and $c_7$ is some universal constant. Then w.p. at least $1 - r_{\max}\exp\left(-c_5 K \log p\right) - \exp\left(-c_6 \log p\right) - c_8 K \exp\left(-c_9(n - r_{\max} - \log p)\right)$, the support union of $\widehat{G}^{(1:K)}$ is the same as that of $\widetilde{G}_0^{(1:K)}$, and that $\left\|\widehat{G}^{(1:K)} - \widetilde{G}^{(1:K)}\right\|_{l_\infty/l_2}/\sqrt{K} = o(g_{\min})$.*

**Discussion on Eq. 11 and Eq. 12.** *(i)* Eq. 11 poses a sample size condition. The value $r_{\max}$ is the sparsity overlap defined in Definition 3.3 (i). It takes value in the interval $[d, \min\{s_0, p, dK\}]$, depending on the similarity in sparsity pattern.

*(ii)* The restriction on $K$ in Eq. 12 plays a similar role as Eq. 10 in Theorem 3.1. This a stronger restriction, but also guarantees the stronger result of support recovery. Existing analyses on $l_1/l_2$-penalized group Lasso were not able to relax this constraint, neither, so some of them treated $K$ as a constant in the analysis [18, 19]. Recall that in Theorem 3.1, we were able to allow $K = \mathcal{O}(p \log p)$. Technically, this was achieved because in the proof of Theorem 3.1, we avoid analyzing the general recovery of group Lasso, but only its null-consistency (i.e., the special case of true structures having zero support), where tighter bound can be derived and it is sufficient for order recovery.

**Benefit of joint estimation.** Eq. 13 plays a similar role as Eq. 8 in Theorem 3.1. It specifies a rate at which the sample size must grow for successful union support recovery. As long as $r_{\max}$ is of order $o(\sqrt{K})$, $K$ will effectively reduce the second term in Eq. 8. Apart from that, the recover probability specified in Theorem 3.2 grows in $K$.

# 4 Algorithm

Solving the optimization in Eq. 7 by searching over all permutations $\pi \in \mathbb{S}_p$ is intractable due to the large combinatorial space. Inspired by the smooth characterization of acyclic graph [16], we propose a continuous optimization problem, whose optimizer is the same as the estimator in Eq. 7. Furthermore, we will design an efficient iterative algorithm to approximate the solution.

## 4.1 Continuous program

We convert Eq. 7 to the following constrained continuous program

$$\min_{\substack{T\in\mathbb{R}^{p\times p} \\ G^{(1)},\cdots,G^{(K)}\in\mathbb{R}^{p\times p}}} \sum_{k=1}^{K} \frac{1}{2n}\left\|\boldsymbol{X}^{(k)} - \boldsymbol{X}^{(k)}\overline{G}^{(k)}\right\|_F^2 + \lambda\|\overline{G}^{(1:K)}\|_{l_1/l_2} + \rho\|\mathbf{1}_{p\times p} - T\|_F^2 \tag{14}$$

$$\text{subject to} \quad h(T) := \mathrm{trace}(e^{T\circ T}) - p = 0, \tag{15}$$

where $\overline{G}^{(k)} := G^{(k)}\circ T$ is element-wise multiplication between $G^{(k)}$ and $T$, and $\mathbf{1}_{p\times p}$ is a $p\times p$ matrix with entries equal to one. Eq. 15 is a smooth 'DAG-ness' constraint proposed by NOTEARS [16], which ensures $T$ is acyclic. One can also use $h(T) := \mathrm{trace}((I + T\circ T)^p) - p$ proposed in [23].

We would like to highlight the novel and interesting design of the matrix $T$ in Eq. 14. What makes Eq. 7 difficult to solve is the requirement that $\{G^{(k)}\}$ must be DAGs and share the same order. A straightforward idea is to apply the smooth acyclic constraint to every $G^{(k)}$, but it is not clear how to enforce their consistent topological order. Our formulation realizes this by a single matrix $T$.

**Algorithm 1:** Joint Estimation Algorithm

---

**Hyperparameters:** $\rho, \alpha, \lambda, t, \delta$
Initialize $G^{(1:K)}, T$ randomly;
**for** $itr = 1, \cdots, M$ **do**

    **for** $itr' = 1, \cdots, M'$ **do**

        $[G^{(1:K)}, T] \leftarrow \texttt{GradOptStep}\big(f; G^{(1:K)}, T, \beta\big);$    ▷ `Gradient-based update on` $f$

        $\forall i, j \in [p], \; G_{ij}^{(1:K)} \leftarrow \frac{G_{ij}^{(1:K)}}{\|G_{ij}^{(1:K)}\|_2} \max\left\{0, \|G_{ij}^{(1:K)}\|_2 - t\lambda|T_{ij}|\right\};$    ▷ `Proximal step`

        $\forall i, j \in [p], \; T_{ij} \leftarrow \text{sign}(T_{ij}) \max\left\{0, |T_{ij}| - t\lambda\|G_{ij}^{(1:K)}\|_2\right\};$    ▷ `Proximal step`

    $\beta \leftarrow \beta + \tau h(T);$                                ▷ `Dual ascent`

    $\alpha \leftarrow \alpha \cdot (1 + \delta);$                         ▷ `Typical rule` [23]

---

To better understand the design rationale of $T$, recall in Eq. 3 that a matrix $G$ is a DAG of order $\pi$ *if and only if* its support set is in $\{(i,j) : \pi(i) < \pi(j)\}$. The matrix $T$ plays a role of restricting the support set of $G^{(k)}$ by masking its entries. Two examples are shown above. However, unlike the learning of masks in other papers which allows $T$ to have any combi-

$$\pi = (1,2,3,4) \qquad\qquad \pi = (4,2,1,3)$$
$$\Updownarrow \qquad\qquad\qquad\qquad \Updownarrow$$
$$T = \begin{bmatrix} 0 & 0 & 0 & 0 \\ 1 & 0 & 0 & 0 \\ 1 & 1 & 0 & 0 \\ 1 & 1 & 1 & 0 \end{bmatrix} \qquad T = \begin{bmatrix} 0 & 0 & 1 & 0 \\ 1 & 0 & 1 & 0 \\ 0 & 0 & 0 & 0 \\ 1 & 1 & 1 & 0 \end{bmatrix}$$

nations of nonzero entries, here we need $T$ to exactly represent the support set for each $\pi$. That is $T$ is from a space $\mathcal{T}_p$ with $p!$ elements: $\mathcal{T}_p := \{T \in \{0,1\}^{p \times p} : T_{ij} = 1 \Leftrightarrow \pi(i) < \pi(j)\}$. Now a key question arises: *How to perform a continuous and differentiable search over $\mathcal{T}_p$?* The following finding motivates our design:

$$T \in \mathcal{T}_p \Longleftrightarrow T \in \arg\min_{T \in \mathbb{R}^{p \times p}} \left\{\|\mathbf{1}_{p \times p} - T\|_F^2 \text{ subject to } h(T) = 0\right\}.$$

In other words, $T$ is a continuous projection of $\mathbf{1}_{p \times p}$ to the space of DAGs. We can then optimize the mask $T$ in the continuous space $\mathbb{R}^{p \times p}$ but the optimal solution must be an element in the discrete space $\mathcal{T}_p$. This observation also naturally leads to the design of Eq. 14.

Finally, we want to emphasize that it is important for the optimal $T$ to have binary entries. Without this property, any nonzero value $c$ can scale the $(G, T)$ pair to give an equivalent masked DAG, i.e., $G \circ T = (cG) \circ (\frac{1}{c}T)$. This scaling equivalence will make the optimization hard to solve in practice.

Proofs for the above arguments and the equivalence between Eq. 7 and Eq. 14 are in Appendix F.

### 4.2 Iterative algorithm

We derive an efficient iterative algorithm using the Lagrangian method with quadratic penalty, which converts Eq. 14 to an unconstrained problem:

$$\min_{T, G^{(1)}, \cdots, G^{(K)} \in \mathbb{R}^{p \times p}} \max_{\beta \geq 0} \mathcal{L}(G^{(1:K)}, T; \beta) := f(G^{(1:K)}, T; \beta) + \lambda \|\overline{G}^{(1:K)}\|_{l_1/l_2},$$

where $f(G^{(1:K)}, T; \beta) := \sum_{k=1}^{K} \frac{1}{2n} \left\|\mathbf{X}^{(k)} - \mathbf{X}^{(k)} \overline{G}^{(k)}\right\|_F^2 + \rho\|\mathbf{1}_{p \times p} - T\|_F^2 + \beta h(T) + \alpha h(T)^2,$

$\beta$ is dual variable, $\alpha$ is the coefficient for quadratic penalty, and $f$ is the smooth term in the objective.

We can solve this min-max problem by alternating primal updates on $(G^{(1:K)}, T)$ and dual updates on $\beta$. Due to the non-smoothness of group norm, the primal update is based on proximal-gradient method, where the **proximal-operator** with respect to $\|\cdot\|_{l_1/l_2}$ has a closed form:

$$\left[\arg\min_{Z^{(1:K)} \in \mathbb{R}^{K \times p \times p}} \frac{1}{2} \sum_{k=1}^{K} \|Z^{(k)} - \mathbf{X}^{(k)}\|_F^2 + c\|Z^{(1:K)}\|_{l_1/l_2}\right]_{ij}^{(1:K)}$$
$$= \frac{\mathbf{X}_{ij}^{(1:K)}}{\|\mathbf{X}_{ij}^{(1:K)}\|_2} \max\{0, \|\mathbf{X}_{ij}^{(1:K)}\|_2 - c\},$$

which is a group-wise soft-threshold. Since $G^{(k)}$ and $T$ are multiplied together element-wisely inside the group norm, the proximal operator will be applied to both of them separately. Together with the

dual update for $\beta$, $\beta \leftarrow \beta + \tau h(T)$ with $\tau$ as the step size, we summarize the overall algorithm for solving Eq. 14 in Algorithm 1.

# 5  Related work

**Single DAG estimation.** Unlike the large literature of research on undirected graphical models [24, 25, 26, 27], statistical guarantees for score-based DAG estimator have been available only in recent years. [14, 15, 17] have shown the DAG estimation consistency in high-dimensions, but they do not consider joint estimation. Nevertheless, some techniques in [14, 15] are useful for our derivations.

**Multi-task learning.** *(i) Undirected graph estimation.* There have been extensive studies on the joint estimation of multiple undirected Gaussian graphical models [28, 29, 30, 31, 32, 33, 34, 35, 36, 37, 38, 39]. *(ii) DAG estimation.* In contrast, not much theoretical work has been done for joint DAG learning. A few pieces of recent works addressed certain statistical aspects of multi-task DAG estimation [9, 20, 21], but [20, 21] tackle the fewer task regime with $K = o(\log p)$, and [9] assumes the causal order is given. Another related work that we notice after the paper submission is [40], which also assumes the DAGs have a consistent causal order, but it focuses on estimating the difference between two DAGs. *(iii) Linear regression.* Multi-task linear regression is also a related topic [41, 42, 43, 44, 45, 46, 47], because the techniques for analyzing group Lasso are used in our analysis [18, 19].

**Practical Algorithms.** Works on practical algorithm design for efficiently solving the score-based optimization are actively conducted [48, 49, 50, 51]. Our algorithm is most related to recent methods exploiting a smooth characterization of acyclicity, including NOTEARS [16] and several subsequent works [23, 52, 53, 54], but they only apply for single-task DAG estimation. Although algorithms for the multi-task counterpart were proposed a decade ago [11, 55, 56, 57], none of them leverage recent advances in characterizing DAGs and providing theoretically guarantees.

# 6  Experiments

## 6.1  Synthetic data

The set of experiments is designed to reveal the effective sample size predicted by Theorem 3.1, and demonstrate the effectiveness of the proposed algorithm. In the simulations, we randomly sample a causal order $\pi$ and a union support set $S_0$. Then we randomly generate multiple DAGs that follow the order $\pi$ and have edges contained in the set $S_0$. For each DAG, we construct a linear SEM with standard Gaussian noise, and sample $n$ data points from it. On tasks with different combinations of $(p, n, s, K)$, we exam the behavior of the joint estimator, estimated by Algorithm 1, on 64 tasks and report the statistics in the following for evaluation. In this experiment, we take $K' = K$ so that all the DAGs are identifiable. This simpler case will make it easier to verify the proposed algorithm and the rates in the theorem.

**Success probability for order recovery.** For each fixed tuple $(p, n, s, K)$, we measure the sample complexity in terms of the parameter $\theta$ specified by Theorem 3.1. Fig 1 plots the success probability $\Pr[\hat{\pi} \in \Pi_0]$, versus $\theta = p/s\sqrt{nK/(p \log p)}$ at a logarithmic scale. Theorem 3.1 predicts that the success probability should transition to 1 once $\theta$ exceeds a critical threshold. Curves in Fig 1 actually have sharp transitions, showing step-function behavior. The sharpness is moderated by the logarithmic scale in $x$-axis. Moreover, by scaling the sample size $n$ using $\theta$, the curves align well as predicted by the theory and have a similar transition point, even though they correspond to very different model dimensions $p$.

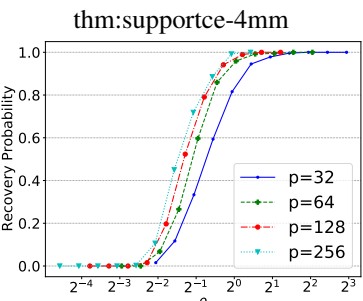

Figure 1: Success probability vs $\log \theta$.

Fig 2 shows the success probability in the form of Heat Maps, where the rows indicate an increase in per task sample size $n$, and the columns indicate an increase in the number of tasks $K$. The results show that the increases in these two quantities have similar effect to the success probability.

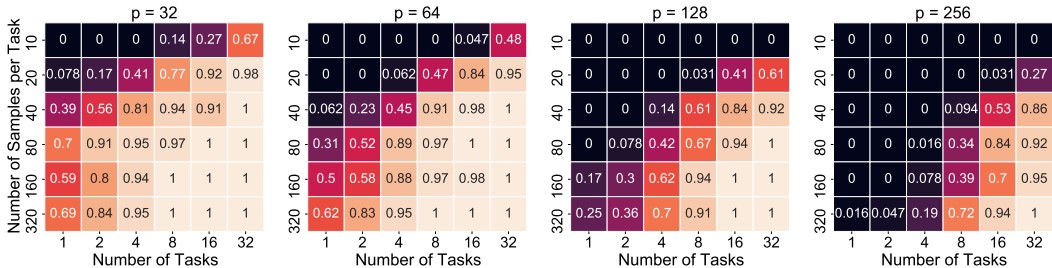

Figure 2: Heat map: Darker colors indicate lower success probability, and lighter colors are higher.

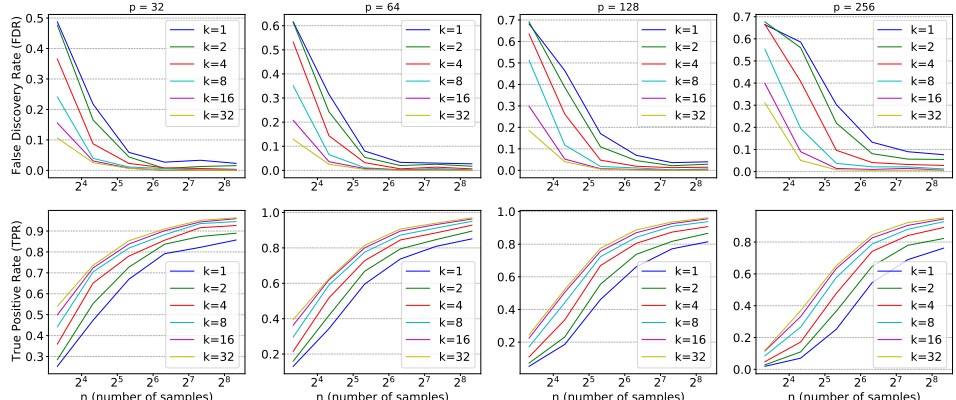

Figure 3: False discovery rate (FDR) and true positive rate (TPR) of the edges.

**Effectiveness of $K$ in structure recovery.** In this experiment, we aim at verifying the effectiveness of the joint estimation algorithm for recovering the structures. For brevity, we only report the numbers for false discovery rate (FDR) and true positive rate (TPR) of edges in Fig 3, but figures for additional metrics can be found in Appendix G. In Fig 3, we can observe consistent improvements when increasing the number of tasks $K$. When per task sample size is small, this improvement reveals to be more obvious.

**Comparison with other joint estimator.** In this experiment, we compare our method MultiDAG with JointGES [21] on models with $p = 32$ and $p = 64$. Results in Table 1 show that two algorithms have similar performance when $K = 1$. However, when $K$ increases, our method returns consistently better structures in terms of SHD.

Table 1: Comparison of MultiDAG (MD) and JointGES (JG) in SHD

|  | $p = 32$ | | | | $p = 64$ | | | |
| --- | --- | --- | --- | --- | --- | --- | --- | --- |
|  | $n = 10$ | $n = 20$ | $n = 80$ | $n = 320$ | $n = 10$ | $n = 20$ | $n = 80$ | $n = 320$ |
| MD(k=1) | $39 \pm 5$ | $25 \pm 5$ | $10 \pm 4$ | $6 \pm 3$ | $104 \pm 7$ | $77 \pm 8$ | $29 \pm 6$ | $19 \pm 8$ |
| MD(k=2) | $37 \pm 5$ | $22 \pm 5$ | $8 \pm 3$ | $4 \pm 3$ | $103 \pm 7$ | $68 \pm 8$ | $22 \pm 6$ | $13 \pm 6$ |
| MD(k=8) | $29 \pm 5$ | $13 \pm 3$ | $4 \pm 2$ | $2 \pm 1$ | $85 \pm 7$ | $42 \pm 6$ | $13 \pm 3$ | $6 \pm 3$ |
| MD(k=32) | $23 \pm 4$ | $11 \pm 3$ | $3 \pm 2$ | $1 \pm 1$ | $66 \pm 5$ | $35 \pm 5$ | $10 \pm 3$ | $3 \pm 2$ |
| JG(k=1) | $31 \pm 5$ | $19 \pm 5$ | $8 \pm 4$ | $6 \pm 5$ | $100 \pm 11$ | $53 \pm 11$ | $18 \pm 11$ | $18 \pm 9$ |
| JG(k=2) | $32 \pm 4$ | $19 \pm 5$ | $9 \pm 5$ | $7 \pm 5$ | $99 \pm 10$ | $51 \pm 10$ | $20 \pm 9$ | $21 \pm 10$ |
| JG(k=8) | $30 \pm 5$ | $19 \pm 5$ | $12 \pm 4$ | $10 \pm 4$ | $82 \pm 10$ | $42 \pm 10$ | $20 \pm 5$ | $27 \pm 9$ |
| JG(k=32) | $26 \pm 4$ | $18 \pm 4$ | $12 \pm 3$ | $9 \pm 3$ | $57 \pm 9$ | $36 \pm 6$ | $19 \pm 5$ | $26 \pm 6$ |

## 6.2 Recovery of gene regulatory network

We investigate how our joint estimator works on more realistic models, by conducting a set of experiments on realistic gene expression data generated by SERGIO [13], which models the additive

effect of cooperative transcription factor binding across multiple gene regulators in parallel with protein degradation and noisy expression.

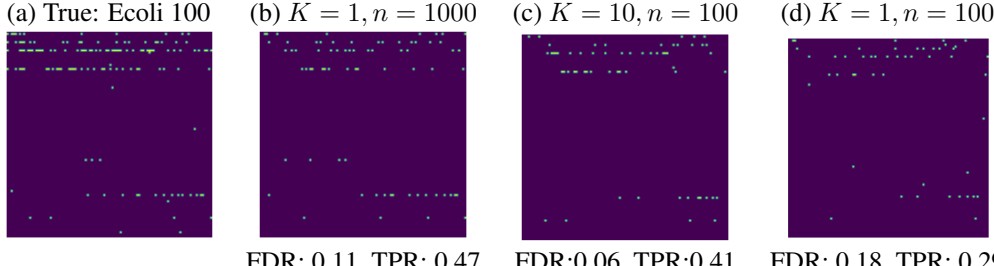

| (a) True: Ecoli 100 | (b) $K = 1, n = 1000$ | (c) $K = 10, n = 100$ | (d) $K = 1, n = 100$ |
|---|---|---|---|
| | FDR: 0.11, TPR: 0.47 | FDR:0.06, TPR:0.41 | FDR: 0.18, TPR: 0.29 |

Figure 4: Visualization of the recovered DAG structures. Each light green colored pixel at position $(i, j)$ indicates an edge from node $i$ to $j$.

We conduct this experiment on the E. coli 100 network, which includes 100 known genes and 137 known regulatory interactions. To evaluate our algorithm, we generate multiple networks by rearranging and re-weighting 5 edges at random in this network without violating the topological order. We simulate the gene expression from each network using SERGIO with a universal non-cooperative hill coefficient of 0.05, which works well with our linear recovery algorithm.

Fig 4 provides a visual comparison of the recovered structures. It can be seem from the true network that there are a few key transcription factors that highlight several rows in the figure. These transcription factors are better identified by the two structures in (b) and (c), but not that clear in (d). Combining this observation with the more quantitative results in Table 2, we see that the combination $(K = 10, n = 100)$ achieves comparable performance to $K = 1$ with the same total number of samples, and outperforms the single task estimation with $n = 100$.

| $K$ | $n$ | FDR | TPR | FPR | SHD |
|---|---|---|---|---|---|
| 1 | 100 | $0.18 \pm 3.8\mathrm{e}{-3}$ | $0.30 \pm 1.2\mathrm{e}{-3}$ | $0.001 \pm 7.3\mathrm{e}{-7}$ | $104.61 \pm 3.2\mathrm{e}1$ |
| 10 | 100 | $0.07 \pm 1.2\mathrm{e}{-3}$ | $0.42 \pm 5.1\mathrm{e}{-4}$ | $0.001 \pm 2.8\mathrm{e}{-7}$ | $84.0 \pm 1.5\mathrm{e}1$ |
| 1 | 1000 | $0.09 \pm 2.9\mathrm{e}{-3}$ | $0.50 \pm 1.4\mathrm{e}{-3}$ | $0.002 \pm 9.2\mathrm{e}{-7}$ | $76.4 \pm 5.9\mathrm{e}1$ |

Table 2: Recovery across 25 independent initializations of SERGIO for each experiment. FPR and SHD stand for false positive rate and structural hamming distance, respectively.

## 7  Conclusion and discussion

In this paper, we have analyzed the behavior of $l_1/l_2$-penalized joint MLE for multiple DAG estimation tasks. Our main result is to show that its performance in recovering the causal order is governed by the sample complexity parameter $\theta(n, K, K', p, s)$ in Eq. 8. Besides, we have proposed an efficient algorithm for approximating the joint estimator via formulating a novel continuous programming, and demonstrated its effectiveness experimentally. The current work applies to DAGs that have certain similarity in sparsity pattern. It will be interesting to consider whether the joint estimation without the group-norm (and without the union support assumption) can also lead to similar improvement in causal order recovery.

## Acknowledgement

Xinshi Chen is supported by the Google PhD Fellowship. This works is done partially during a visit at MBZUAI. We are grateful for the computing resources provided by the Partnership for an Advanced Computing Environment (PACE) at the Georgia Institute of Technology, Atlanta. We are thankful for PACE Research Scientist Fang (Cherry) Liu's excellent HPC consulting.

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
