## A List of definitions and notations

For the convenience of the reader, we summarize a list of notations blow.

1. $\widehat{G}_j(\hat{\pi}) := [\hat{g}_j^{(1)}(\hat{\pi}), \cdots, \hat{g}_j^{(K)}(\hat{\pi})]$ and $\widetilde{G}_j(\hat{\pi}) := [\tilde{g}_j^{(1)}(\hat{\pi}), \cdots, \tilde{g}_j^{(K)}(\hat{\pi})]$.

2. For all $k \in [K]$, the eigenvalues of $\Sigma^{(k)}$ are in $[\Lambda_{\min}, \Lambda_{\max}]$, for some constants $0 < \Lambda_{\min} \leq \Lambda_{\max} < \infty$.

3. $\sigma_{\max} = \sup_{k\in[K],j\in[p],\pi\in\mathbb{S}_p} |\sigma_j^{(k)}(\pi)|$. Note that $\sigma_{\max}^2 \leq \Lambda_{\max}$.

4. $c_{\max} := \sup_{k,j,\pi} |1 - \sigma_j^{(k)}(\pi)^2|$. Note that $c_{\max} \leq 1 + \sigma_{\max}^2 \leq 1 + \Lambda_{\max}$.

5. $\rho = \sup_{k\in[K],j\in[p]} \Sigma_{jj}^{(k)}$. Note that $\rho \leq \Lambda_{\max}$.

6. $s(\pi) := |S(\pi)|$ where $S(\pi) := \cup_{k\in[K]}\text{supp}(\widetilde{G}^{(k)}(\pi))$. $s_0 := s(\pi_0)$. $s := \sup_{\pi\in\mathbb{S}_p} s(\pi)$.

7. $g_{\max} := \sup_{\pi\in\mathbb{S}_p,(i,j)\in S(\pi)} \left\| \widetilde{G}_{ij}^{(1:K)}(\pi) \right\|_2 / \sqrt{K}$.

8. $g_{\min} := \inf_{(i,j)\in S(\pi_0)} \left\| \widetilde{G}_{0ij}^{(1:K)} \right\|_2 / \sqrt{K}$.

9. $\text{RS}_j := \{i \in [p] : \exists k \in [K] \ s.t. \ \widetilde{G}_{0ij}^{(k)} \neq 0\}$.

10. $r_{\max} := \sup_{j\in[p]} \left|\text{RS}_j\right|$.

11. $D_{\max} := \sup_{j\in[p],S=\text{RS}_j,k\in[K]} \left\| (\Sigma_{SS}^{(k)})^{-1} \right\|_\infty$.

12. $\rho_u := \sup_{j\in[p],S=\text{RS}_j,k\in[K]} \max_{i\in S^c} \left(\Sigma_{S^cS^c|S}^{(k)}\right)_{ii}$.

13. $S_j(\pi) := \{i : \pi(i) < \pi(j)\}$.

14. $U_j(\pi) := \cup_{k\in[K]}\text{supp}(\tilde{g}_j^{(k)}(\pi)) = \{i : \exists k \in [K] \ s.t. \ \widetilde{G}_{ij}^{(k)}(\pi) \neq 0\}$.

15. $d_j := \sup_{\pi\in\mathbb{S}_p} |U_j(\pi)|$. $d = \sup_{j\in[p]} d_j$.

## B Details of Theorem 3.1: causal order recovery

In Appendix B.1, we present a general statement of Theorem 3.1 (a) along with its proof. Proof of part (b) in Theorem 3.1 is given in Appendix B.3.

### B.1 Order recovery: proof of Theorem 3.1 (a) (Theorem B.1)

Theorem 3.1 (a) states the order recovery guarantee for a specified parameter $\lambda = \sqrt{\frac{p \log p}{n}}$. In the following, we will present a more general statement of Theorem 3.1 (a) that does not specify the choice of $\lambda$, after which we will present the proof.

**Theorem B.1** (General statement of Theorem 3.1 (a)). *For any $\delta_1, \delta_2, \delta_3, \delta_4, \delta_5 \in (0,1)$, if the following conditions are satisfied*

$$K \leq \frac{\delta_2^2\delta_3^2\delta_4^2\lambda^2 n}{64\rho\sigma_{\max}^2},$$

$$n \geq \left((4(1-\delta_3)\delta_3^{-2} - \delta_5)\right)^{-1} (\log K + (d+1)\log p),$$

$$\frac{1}{\eta_w} > \left(\frac{16\sigma_{\max}^8}{\delta_1(1-\delta_1)} + \frac{4\sigma_{\max}^4 c_{\max}}{1-\delta_1}\right) \frac{2\log p}{nK'} + \frac{4\sigma_{\max}^4 g_{\max}}{\delta_1} \frac{\lambda\left(s(\pi_0) + \delta_2 s(\hat{\pi})\right)}{p}\sqrt{\frac{K}{K'^2}}$$

$$+ \frac{8\sigma_{\max}^6}{\delta_1}\sqrt{\frac{2\log p}{n}}\frac{K-K'}{K'^2},$$

*then $\hat{\pi} = \pi_0$ with probability at least*

$$1 - \exp\left(-t^*(1-\delta_4) + (d+2)\log p\right) - \exp\left(-\delta_5 n\right) - 2\exp\left(-p\log p\right),$$

*where $t^* := \frac{\delta_2^2\delta_3^2\lambda^2 n}{16\rho\sigma_{\max}^2}$.*

**Proof outline.** By optimality of the joint estimator, (for simplicity, we write $\widehat{G}^{(k)} := \widehat{G}^{(k)}(\hat{\pi})$)

$$\sum_{k=1}^{K} \frac{1}{2n} \|\boldsymbol{X}^{(k)}\widehat{G}^{(k)} - \boldsymbol{X}^{(k)}\widetilde{G}^{(k)}(\hat{\pi})\|_F^2 + \lambda \|\widehat{G}^{(1:K)}\|_{l_1/l_2}$$

$$\leq \underbrace{\sum_{k=1}^{K} \frac{1}{2n} \left( \|\boldsymbol{X}^{(k)} - \boldsymbol{X}^{(k)}\widetilde{G}_0^{(k)}\|_F^2 - \|\boldsymbol{X}^{(k)} - \boldsymbol{X}^{(k)}\widetilde{G}^{(k)}(\hat{\pi})\|_F^2 \right)}_{(I)}$$

$$+ \underbrace{\sum_{k=1}^{K} \frac{1}{n} \langle \boldsymbol{X}^{(k)} - \boldsymbol{X}^{(k)}\widetilde{G}^{(k)}(\hat{\pi}), \boldsymbol{X}^{(k)}\widehat{G}^{(k)} - \boldsymbol{X}^{(k)}\widetilde{G}^{(k)}(\hat{\pi}) \rangle_F}_{(II)} + \lambda \|\widetilde{G}_0^{(1:K)}\|_{l_1/l_2}.$$

The proof is based on a bound for the term (I) and a bound for the term (II).

To bound (I), we show that the empirical variances of the error terms are close to their expectations, which is achieved mainly by a concentration bound on a linear combination of Chi-squared random variables.

To bound (II), we show the following inequality holds true for all $j \in [p]$ and $\hat{\pi} \in \mathbb{S}_p$ with high probability (where $\widetilde{\varepsilon}_j^{(k)}(\hat{\pi})$ is the empirical error):

$$\sup_{\{\beta^{(k)} \in \mathbb{R}^m\}} \frac{1}{n} \sum_{k=1}^{K} \langle \widetilde{\varepsilon}_j^{(k)}(\hat{\pi}), \boldsymbol{X}_{S_j}^{(k)}\beta^{(k)} \rangle - \frac{\delta}{2n} \sum_{k=1}^{K} \|\boldsymbol{X}_{S_j}^{(k)}\beta^{(k)}\|_2^2 - \delta\lambda \|[\beta^{(1)}, \cdots, \beta^{(K)}]\|_{l_1/l_2} \leq 0.$$

We highlight two technical aspects in bounding (II):

- For each fixed $j$ and $\hat{\pi}$, the above inequality is proved by showing the **null-consistency** of $l_1/l_2$-penalized group Lasso problem (see Appendix B.2.3). Null-consistency means successfully recovering the true linear regression model when the true parameters have null support (all parameters are zeros). Technically, the improvement in sample complexity for recovering multiple DAGs partially comes from the benefit of a larger $K$ for guaranteeing the null-consistency.

- We need to insure the bounds hold uniformly over all permutations $\hat{\pi} \in \mathbb{S}_p$ and $j \in [p]$. To avoid using a naive union bound over $p!$ many permutations, we leverage the sparsity of the graph structures and prove that the number of elements in the set $\{\widetilde{G}^{(1:K)}(\pi) : \pi \in \mathbb{S}_p\}$ can be fewer than $p!$ (see Appendix E), so that we can take a uniform control over this smaller set instead.

We summarize the bounds for (I) and (II) in Lemma B.1 and Lemma B.2, which can be found in Appendix B.2.1 and Appendix B.2.2.

**Detailed proof of Theorem B.1.** Collecting the results in Lemma B.1 and Lemma B.2 and reorganizing the terms in the inequalities, we have the following conclusion.

For any $\delta_1, \delta_2, \delta_3, \delta_4, \delta_5 \in (0,1)$ and $t^* := \frac{\delta_2^2 \delta_3^2 \lambda^2 n}{16\rho\sigma_{\max}^2}$, if the following conditions are satisfied:

$$K \leq \frac{\delta_4^2}{4} t^* = \frac{\delta_2^2 \delta_3^2 \delta_4^2 \lambda^2 n}{64\rho\sigma_{\max}^2}$$

$$\left(4(1-\delta_3)\delta_3^{-2} - \delta_5\right) n \geq \log K + (d+1)\log p,$$

then with probability at least $1 - \exp\left(-t^*(1 - \delta_4) + (d+2)\log p\right) - \exp\left(-\delta_5 n\right) - 2\exp\left(-p\log p\right)$, it holds for all $\hat{\pi} \in \mathbb{S}_p$ that

$$\frac{1 - \delta_2}{2n} \sum_{k=1}^{K} \|\boldsymbol{X}^{(k)}\widehat{G}^{(k)} - \boldsymbol{X}^{(k)}\widetilde{G}^{(k)}(\hat{\pi})\|_F^2 + \lambda\|\widehat{G}^{(1:K)}\|_{l_1/l_2}$$

$$+ \frac{\delta_1}{4\sigma_{\max}^4} \sum_{k=1}^{K'} \sum_{j=1}^{p} \left(\sigma_j^{(k)}(\hat{\pi})^2 - \sigma_j^{(k)}(\pi_0)^2\right)^2$$

$$\leq \left(\frac{4\sigma_{\max}^4}{1 - \delta_1} + 2\sigma_{\max}^2\right)\frac{2p\log p}{n} + 2\sigma_{\max}^2 \sqrt{\frac{(K - K')2p^2\log p}{n}} \tag{16}$$

$$+ \delta_2\lambda\|\widehat{G}^{(1:K)} - \widetilde{G}^{(1:K)}(\hat{\pi})\|_{l_1/l_2} + \lambda\|\widetilde{G}_0^{(1:K)}\|_{l_1/l_2}$$

$$\leq \left(\frac{4\sigma_{\max}^4}{1 - \delta_1} + 2\sigma_{\max}^2\right)\frac{2p\log p}{n} + 2\sigma_{\max}^2 \sqrt{\frac{(K - K')2p^2\log p}{n}}$$

$$+ \delta_2\lambda\|\widehat{G}^{(1:K)}\|_{l_1/l_2} + \delta_2\lambda\|\widetilde{G}^{(1:K)}(\hat{\pi})\|_{l_1/l_2} + \lambda\|\widetilde{G}_0^{(1:K)}\|_{l_1/l_2}. \tag{17}$$

Suppose $\hat{\pi} \neq \pi_0$. Condition 3.5 implies

$$\frac{\delta_1}{4\sigma_{\max}^4}\frac{pK'}{\eta_w} \leq \left(\frac{4\sigma_{\max}^4}{1 - \delta_1} + 2\sigma_{\max}^2\right)\frac{2p\log p}{n} + \lambda\|\widetilde{G}_0^{(1:K)}\|_{l_1/l_2} + \delta_2\lambda\|\widetilde{G}^{(1:K)}(\hat{\pi})\|_{l_1/l_2}$$

$$+ 2\sigma_{\max}^2\sqrt{\frac{(K - K')2p^2\log p}{n}}.$$

Divide both sides by $pK'$, it implies

$$\frac{\delta_1}{4\sigma_{\max}^4}\frac{1}{\eta_w} \leq \left(\frac{4\sigma_{\max}^4}{1 - \delta_1} + 2\sigma_{\max}^2\right)\frac{2\log p}{nK'} + \frac{\lambda\|\widetilde{G}_0^{(1:K)}\|_{l_1/l_2}}{pK'} + \delta_2\frac{\lambda\|\widetilde{G}^{(1:K)}(\hat{\pi})\|_{l_1/l_2}}{pK'}$$

$$+ 2\sigma_{\max}^2\sqrt{\frac{(K - K')2\log p}{nK'^2}}$$

$$\leq \left(\frac{4\sigma_{\max}^4}{1 - \delta_1} + c_{\max}\right)\frac{2\log p}{nK'} + \frac{\lambda\left(s(\pi_0) + \delta_2 s(\hat{\pi})\right)g_{\max}\sqrt{K}}{pK'}$$

$$+ 2\sigma_{\max}^2\sqrt{\frac{(K - K')2\log p}{nK'^2}}.$$

The last inequality uses the fact that $\|\widetilde{G}^{(1:K)}(\pi)\|_{l_1/l_2} \leq s(\pi)\sqrt{K}g_{\max}$. It contradicts with the condition

$$\frac{1}{\eta_w} > \left(\frac{16\sigma_{\max}^8}{\delta_1(1 - \delta_1)} + \frac{4\sigma_{\max}^4 c_{\max}}{1 - \delta_1}\right)\frac{2\log p}{nK'} + \frac{4\sigma_{\max}^4 g_{\max}}{\delta_1}\frac{\lambda\left(s(\pi_0) + \delta_2 s(\hat{\pi})\right)}{p}\sqrt{\frac{K}{K'^2}}$$

$$+ \frac{8\sigma_{\max}^6}{\delta_1}\sqrt{\frac{2\log p}{n}\frac{K - K'}{K'^2}}.$$

Therefore, $\hat{\pi} \in \Pi_0$.

Theorem 3.1 (a) is straightforward by taking $\lambda = \sqrt{p\log p/n}$.

## B.2 Key lemmas for proving Theorem 3.1 (a)

### B.2.1 Lemma B.1: Analysis of (I)

**Lemma B.1.** *Denote* $\sigma_{\max} = \sup_{k\in[K], j\in[p], \pi\in\mathbb{S}_p} |\sigma_j^{(k)}(\pi)|$. *With probability at least* $1 - 2e^{-p\log p}$, *it holds for any* $\delta_1 \in (0,1)$ *and any permutations* $\hat{\pi} \in \mathbb{S}_p$ *that,*

$$
\sum_{k=1}^{K} \frac{1}{2n} \left( \|\boldsymbol{X}^{(k)} - \boldsymbol{X}^{(k)}\widetilde{G}_0^{(k)}\|_F^2 - \|\boldsymbol{X}^{(k)} - \boldsymbol{X}^{(k)}\widetilde{G}^{(k)}(\hat{\pi})\|_F^2 \right)
$$

$$
\leq - \frac{\delta_1}{4\sigma_{\max}^4} \sum_{k=1}^{K'} \sum_{j=1}^{p} \left( \sigma_j^{(k)}(\pi_0)^2 - \sigma_j^{(k)}(\hat{\pi})^2 \right)^2
$$

$$
+ \left( \frac{\sigma_{\max}^4}{1 - \delta_1} + 2\sigma_{\max}^2 \right) \frac{2p \log p}{n} + 2\sigma_{\max}^2 \sqrt{\frac{(K - K')2p^2 \log p}{n}}.
$$

We now state the proof of this Lemma. Denote the $j$-th column of $\widetilde{G}^{(k)}(\pi)$ as $\widetilde{g}_j^{(k)}(\pi)$, and the noise as

$$
\widetilde{\varepsilon}_j^{(k)}(\pi) := \boldsymbol{X}_j^{(k)} - \boldsymbol{X}^{(k)}\widetilde{g}_j^{(k)}(\pi) \in \mathbb{R}^n. \tag{18}
$$

Then we can rewrite the term (I) as follows.

$$
(I) = \sum_{k=1}^{K} \frac{1}{2n} \left( \|\boldsymbol{X}^{(k)} - \boldsymbol{X}^{(k)}\widetilde{G}_0^{(k)}\|_F^2 - \|\boldsymbol{X}^{(k)} - \boldsymbol{X}^{(k)}\widetilde{G}^{(k)}(\hat{\pi})\|_F^2 \right)
$$

$$
= \frac{1}{2} \sum_{k=1}^{K} \left[ \sum_{j=1}^{p} \frac{\frac{1}{n}\|\widetilde{\varepsilon}_j^{(k)}(\hat{\pi})\|_2^2}{\sigma_j^{(k)}(\hat{\pi})^2} \sigma_j^{(k)}(\pi_0)^2 - \sum_{j=1}^{p} \frac{1}{n}\|\widetilde{\varepsilon}_j^{(k)}(\hat{\pi})\|_2^2 \right]
$$

$$
= \frac{1}{2} \sum_{k=1}^{K} \sum_{j=1}^{p} \left( \sigma_j^{(k)}(\pi_0)^2 - \sigma_j^{(k)}(\hat{\pi})^2 \right) \left( \frac{\frac{1}{n}\|\widetilde{\varepsilon}_j^{(k)}(\hat{\pi})\|_2^2}{\sigma_j^{(k)}(\hat{\pi})^2} - 1 \right) + \frac{1}{2} \sum_{k=1}^{K} \sum_{j=1}^{p} \left( \sigma_j^{(k)}(\pi_0)^2 - \sigma_j^{(k)}(\hat{\pi})^2 \right)
$$

$$
\leq \frac{1}{2} \sum_{k=1}^{K} \sum_{j=1}^{p} \left( \sigma_j^{(k)}(\pi_0)^2 - \sigma_j^{(k)}(\hat{\pi})^2 \right) \left( \frac{\frac{1}{n}\|\widetilde{\varepsilon}_j^{(k)}(\hat{\pi})\|_2^2}{\sigma_j^{(k)}(\hat{\pi})^2} - 1 \right)
$$

$$
- \frac{1}{4\sigma_{\max}^4} \sum_{k=1}^{K'} \sum_{j=1}^{p} \left( \sigma_j^{(k)}(\hat{\pi})^2 - \sigma_j^{(k)}(\pi_0)^2 \right)^2
$$

The last inequality holds because for $k = 1, \cdots, K'$, $\sum_{j=1}^{p} \left( \sigma_j^{(k)}(\pi_0)^2 - \sigma_j^{(k)}(\hat{\pi})^2 \right) \leq -\frac{1}{2\sigma_{\max}^4} \sum_{j=1}^{p} \left( \sigma_j^{(k)}(\hat{\pi})^2 - \sigma_j^{(k)}(\pi_0)^2 \right)^2$, and that for $k > K'$, $\sum_{j=1}^{p} \left( \sigma_j^{(k)}(\pi_0)^2 - \sigma_j^{(k)}(\hat{\pi})^2 \right) \leq 0$. Then we bound the first term using the concentration bound on Chi-squared random variables.

$$
\sum_{k=1}^{K} \sum_{j=1}^{p} \left( \sigma_j^{(k)}(\pi_0)^2 - \sigma_j^{(k)}(\hat{\pi})^2 \right) \left( \frac{\frac{1}{n}\|\widetilde{\varepsilon}_j^{(k)}(\hat{\pi})\|_2^2}{\sigma_j^{(k)}(\hat{\pi})^2} - 1 \right)
$$

$$
\overset{d.}{=} \sum_{k=1}^{K} \sum_{j=1}^{p} \left( \sigma_j^{(k)}(\pi_0)^2 - \sigma_j^{(k)}(\hat{\pi})^2 \right) \left( \frac{1}{n}\xi_j^2 - 1 \right) = \frac{1}{n} \sum_{k=1}^{K} \sum_{j=1}^{p} \left( \sigma_j^{(k)}(\pi_0)^2 - \sigma_j^{(k)}(\hat{\pi})^2 \right) \left( \xi_j^2 - n \right),
$$

where $\xi_j^2 \sim \chi^2(n)$ are i.i.d. Chi-squared random variables of degree $n$.

By Lemma H.1, for any fixed $\hat{\pi} \in \mathbb{S}_p$ and for any $t > 0$, it holds with probability at least $1 - e^{-t}$ that

$$\frac{1}{n} \sum_{k=1}^{K'} \sum_{j=1}^{p} \left( \sigma_j^{(k)}(\pi_0)^2 - \sigma_j^{(k)}(\hat{\pi})^2 \right) \left( \xi_j^2 - n \right)$$

$$\leq 2 \sqrt{\frac{\sum_{k=1}^{K'} \sum_{j=1}^{p} \left( \sigma_j^{(k)}(\pi_0)^2 - \sigma_j^{(k)}(\hat{\pi})^2 \right)^2}{n}} \sqrt{t} + \frac{2\sigma_{\max}^2}{n} t$$

$$\leq \delta \sum_{k=1}^{K'} \sum_{j=1}^{p} \left( \sigma_j^{(k)}(\pi_0)^2 - \sigma_j^{(k)}(\hat{\pi})^2 \right)^2 + \left( \frac{1}{\delta} + 2\sigma_{\max}^2 \right) \frac{t}{n},$$

The last inequality holds because $2ab \leq \delta a^2 + \frac{1}{\delta} b^2$ for any $\delta > 0$. Now it remains to take a union bound over the permutation $\hat{\pi} \in \mathbb{S}_p$. There are $p!$ many permutations. Take an uniform control over all possible $\hat{\pi} \in \mathbb{S}_p$. It implies that with probability at least $1 - (p!)e^{-t}$, the above inequality holds for all $\hat{\pi} \in \mathbb{S}_p$. Equivalently, we can say it holds with probability at least $1 - e^{-t}$ that it holds for all $\hat{\pi}$ that

$$\frac{1}{n} \sum_{k=1}^{K'} \sum_{j=1}^{p} \left( \sigma_j^{(k)}(\pi_0)^2 - \sigma_j^{(k)}(\hat{\pi})^2 \right) \left( \xi_j^2 - n \right)$$

$$\leq \delta \sum_{k=1}^{K'} \sum_{j=1}^{p} \left( \sigma_j^{(k)}(\pi_0)^2 - \sigma_j^{(k)}(\hat{\pi})^2 \right)^2 + \left( \frac{1}{\delta} + 2\sigma_{\max}^2 \right) \frac{t + p \log p}{n}.$$

For the non-identifiable models, we can use Lemma H.1 in a similar way to obtain that with probability at least $1 - e^{-t}$, the following holds for all $\hat{\pi}$,

$$\frac{1}{n} \sum_{k=K'+1}^{K} \sum_{j=1}^{p} \left( \sigma_j^{(k)}(\pi_0)^2 - \sigma_j^{(k)}(\hat{\pi})^2 \right) \left( \xi_j^2 - n \right)$$

$$\leq 2 \sqrt{\frac{\sum_{k=K'+1}^{K} \sum_{j=1}^{p} \left( \sigma_j^{(k)}(\pi_0)^2 - \sigma_j^{(k)}(\hat{\pi})^2 \right)^2}{n}} \sqrt{t} + \frac{2\sigma_{\max}^2}{n} t$$

$$\leq 2\sigma_{\max}^2 \sqrt{\frac{(K - K')p(t + p \log p)}{n}} + \frac{2\sigma_{\max}^2}{n}(t + p \log p).$$

Putting the above results back into the term (I), taking $\delta' = 2\delta$, and taking $t = p \log p$, we have with probability at least $1 - 2e^{-p \log p}$ that

$$(I) \leq \frac{\delta'}{4} \sum_{k=1}^{K'} \sum_{j=1}^{p} \left( \sigma_j^{(k)}(\pi_0)^2 - \sigma_j^{(k)}(\hat{\pi})^2 \right)^2 - \frac{1}{4\sigma_{\max}^4} \sum_{k=1}^{K'} \sum_{j=1}^{p} \left( \sigma_j^{(k)}(\hat{\pi})^2 - \sigma_j^{(k)}(\pi_0)^2 \right)^2$$

$$+ \left( \frac{1}{\delta'} + 2\sigma_{\max}^2 \right) \frac{2p \log p}{n} + 2\sigma_{\max}^2 \sqrt{\frac{(K - K')2p^2 \log p}{n}}$$

Finally, take $\delta_1 = 1 - \sigma_{\max}^4 \delta'$ so that $\delta' = \frac{1}{\sigma_{\max}^4}(1 - \delta_1)$. Then for any $\delta_1 \in (0, 1)$, the following inequality holds with probability at least $1 - e^{-p \log p}$ for all $\hat{\pi} \in \mathbb{S}_p$:

$$(I) \leq -\frac{\delta_1}{4\sigma_{\max}^4} \sum_{k=1}^{K'} \sum_{j=1}^{p} \left( \sigma_j^{(k)}(\pi_0)^2 - \sigma_j^{(k)}(\hat{\pi})^2 \right)^2$$

$$+ \left( \frac{\sigma_{\max}^4}{1 - \delta_1} + 2\sigma_{\max}^2 \right) \frac{2p \log p}{n} + 2\sigma_{\max}^2 \sqrt{\frac{(K - K')2p^2 \log p}{n}}.$$

### B.2.2  Lemma B.2: Analysis of (II)

**Lemma B.2.** *Denote $\rho := \sup_{k\in[K],j\in[p]} \Sigma_{jj}^{(k)}$ and $\sigma_{\max} := \sup_{k\in[K],j\in[p],\pi\in\mathbb{S}_p} |\sigma_j^{(k)}(\pi)|$. For any $\delta_2, \delta_3 \in (0,1)$, assume $\lambda$ satisfies $t^* := \frac{\delta_2^2\delta_3^2\lambda^2 n}{16\rho\sigma_{\max}^2} > K$. With probability at least*

$$1 - \exp\left(-t^*\left[1 - 2\sqrt{\frac{K}{t^*}}\right] + (d+2)\log p\right) - \exp\left(-\frac{4(1-\delta_3)}{\delta_3^2}n + \log K + (d+1)\log p\right),$$

*the following inequality holds true:*

$$\frac{1}{2n}\sum_{k=1}^K \langle \boldsymbol{X}^{(k)} - \boldsymbol{X}^{(k)}\widetilde{G}^{(k)}(\hat{\pi}), \boldsymbol{X}^{(k)}\widehat{G}^{(k)} - \boldsymbol{X}^{(k)}\widetilde{G}^{(k)}(\hat{\pi})\rangle$$

$$\leq \frac{\delta_2}{2n}\sum_{k=1}^K \|\boldsymbol{X}^{(k)}\widehat{G}^{(k)} - \boldsymbol{X}^{(k)}\widetilde{G}^{(k)}(\hat{\pi})\|_F^2 + \delta_2\lambda\|\widehat{G}^{(1:K)} - \widetilde{G}^{(1:K)}(\hat{\pi})\|_{l_1/l_2}.$$

We now state the proof of this Lemma. To show the inequality in Lemma B.2 holds true, it is sufficient to show the following inequality holds true for all $j$ and $\hat{\pi}$:

$$\frac{1}{n}\sum_{k=1}^K \langle \widetilde{\varepsilon}_j^{(k)}(\hat{\pi}), \boldsymbol{X}^{(k)}\left(\widehat{g}_j^{(k)}(\hat{\pi}) - \widetilde{g}_j^{(k)}(\hat{\pi})\right)\rangle_F \leq \frac{\delta}{2n}\sum_{k=1}^K \|\boldsymbol{X}^{(k)}\left(\widehat{g}_j^{(k)}(\hat{\pi}) - \widetilde{g}_j^{(k)}(\hat{\pi})\right)\|_2^2$$

$$- \delta\lambda\|\widehat{g}_j^{(1:K)}(\hat{\pi}) - \widetilde{g}_j^{(1:K)}(\hat{\pi})\|_{l_1/l_2}, \qquad (19)$$

where we denote

$$\widehat{G}_j(\hat{\pi}) := [\widehat{g}_j^{(1)}(\hat{\pi}), \cdots, \widehat{g}_j^{(K)}(\hat{\pi})] \quad \text{and} \quad \widetilde{G}_j(\hat{\pi}) := [\widetilde{g}_j^{(1)}(\hat{\pi}), \cdots, \widetilde{g}_j^{(K)}(\hat{\pi})].$$

Now consider a fixed $j$ and a fixed $\hat{\pi}$. Recall $S_j(\hat{\pi})$ which denotes the set of ancestors of the node $j$ specified by the permutation $\hat{\pi}$, and let $m = |S_j(\hat{\pi})| \in [0, p-1]$ be its cardinality. Let $\boldsymbol{X}_{S_j}^{(k)} = \boldsymbol{X}^{(k)}|_{S_j(\hat{\pi})}$ denote the submatrix of $\boldsymbol{X}^{(k)}$ whose column indices are in $S_j(\hat{\pi})$. We define the event

$$\mathcal{E}(\delta, \lambda; \widetilde{\varepsilon}_j^{(k)}(\hat{\pi})) :=$$

$$\left\{ \sup_{\{\beta^{(k)}\in\mathbb{R}^m\}} \frac{1}{n}\sum_{k=1}^K \langle \widetilde{\varepsilon}_j^{(k)}(\hat{\pi}), \boldsymbol{X}_{S_j}^{(k)}\beta^{(k)}\rangle - \frac{\delta}{2n}\sum_{k=1}^K \|\boldsymbol{X}_{S_j}^{(k)}\beta^{(k)}\|_2^2 - \delta\lambda\|[\beta^{(1)}, \cdots, \beta^{(K)}]\|_{l_1/l_2} \leq 0 \right\}.$$

It's easy to see that with probability at least $\Pr\left[\mathcal{E}(\delta, \lambda; \widetilde{\varepsilon}_j^{(1:K)}(\hat{\pi}))\right]$, the inequality in Eq. 19 holds true. Therefore, we need to derive the probability of the joint event $\cap_{j\in[p],\hat{\pi}\in\mathbb{S}_p}\mathcal{E}(\delta, \lambda; \widetilde{\varepsilon}_j^{(1:K)}(\hat{\pi}))$ in this proof. Observe that:

$$\mathcal{E}(\delta, \lambda; \widetilde{\varepsilon}_j^{(1:K)}(\hat{\pi})) \subseteq$$

$$\left\{ \sup_{\{\beta^{(k)}\in\mathbb{R}^m\}} \frac{1}{2n}\sum_{k=1}^K \|\frac{\widetilde{\varepsilon}_j^{(k)}(\hat{\pi})}{\delta}\|_2^2 - \frac{1}{2n}\sum_{k=1}^K \|\frac{\widetilde{\varepsilon}_j^{(k)}(\hat{\pi})}{\delta} - \boldsymbol{X}_{S_j}^{(k)}\beta^{(k)}\|_2^2 - \lambda\|[\beta^{(1)}, \cdots, \beta^{(K)}]\|_{l_1/l_2} \leq 0 \right\}$$

$$= \left\{ \boldsymbol{0} \in \arg\min_{\{\beta^{(k)}\in\mathbb{R}^m\}} \frac{1}{2n}\sum_{k=1}^K \|\frac{\widetilde{\varepsilon}_j^{(k)}(\hat{\pi})}{\delta} - \boldsymbol{X}_{S_j}^{(k)}\beta^{(k)}\|_2^2 + \lambda\|[\beta^{(1)}, \cdots, \beta^{(K)}]\|_{l_1/l_2} \right\}$$

Therefore, we resort to bound the probability of the above event, which is the **null-consistency** of $l_1/l_2$-penalized group Lasso problem. We present the null-consistency analysis by Lemma B.3, and its proof is given in Sec B.2.3.

Note that in the event $\mathcal{E}(\delta, \lambda; \widetilde{\varepsilon}_j^{(1:K)}(\hat{\pi}))$, the variance is $\frac{\widetilde{\varepsilon}_j^{(k)}(\hat{\pi})}{\delta} \overset{d.}{=} \boldsymbol{w}_k$ with

$$\boldsymbol{w}_k \sim \mathcal{N}\left(0, \left(\frac{\sigma_j^{(k)}(\hat{\pi})}{\delta}\right)^2 I_n\right).$$

Therefore, we can take $\sigma_0$ in Lemma B.3 to be $\sigma_0 = \frac{\sigma_{\max}}{\delta}$, which implies for any $\delta' \in (0, 1)$, if

$$t^* := \frac{\delta'^2 \delta^2 \lambda^2 n}{16\rho\sigma_{\max}^2} > K,$$

then

$$\Pr\left[ \mathcal{E}(\delta, \lambda; \widetilde{\varepsilon}_j^{(1:K)}(\hat{\pi})) \right] \geq 1 - p\exp\left( -t^*\left[ 1 - 2\sqrt{\frac{K}{t^*}} \right] \right) - K\exp\left( -\frac{4(1-\delta')}{\delta'^2}n \right).$$

Now what remains is to take a uniform control over all events $\cap_{j\in[p], \hat{\pi}\in\mathbb{S}_p} \mathcal{E}(\delta, \lambda; \widetilde{\varepsilon}_j^{(1:K)}(\hat{\pi}))$. A naive way is to enumerate over all permutations $\hat{\pi}$ and all $j$ which will constitute $p \cdot p!$ events. However, recall $\widetilde{\varepsilon}_j^{(k)}(\pi) := \boldsymbol{X}_j^{(k)} - \boldsymbol{X}^{(k)}\widetilde{g}_j^{(k)}(\pi)$. Then it is enough to take a uniform control over the set $\{\widetilde{g}_j^{(1:K)}(\pi) : \pi \in \mathbb{S}_p, j \in [p]\}$. By Eq. 35, this set contains at most $p \cdot p^d$ many elements. Therefore,

$$\Pr\left[ \cap_{j\in[p], \hat{\pi}\in\mathbb{S}_p} \mathcal{E}(\delta, \lambda; \widetilde{\varepsilon}_j^{(1:K)}(\hat{\pi})) \right]$$

$$\geq 1 - p^2 p^d \exp\left( -t^*\left[ 1 - 2\sqrt{\frac{K}{t^*}} \right] \right) - Kp \cdot p^d \exp\left( -\frac{4(1-\delta')}{\delta'^2}n \right)$$

$$\geq 1 - \exp\left( -t^*\left[ 1 - 2\sqrt{\frac{K}{t^*}} \right] + (d+2)\log p \right) - \exp\left( -\frac{4(1-\delta')}{\delta'^2}n + \log K + (d+1)\log p \right)$$

which implies Eq. 19 holds with the above probability.

### B.2.3 Lemma B.3: Null Consistency

**Lemma B.3** (Null-consistency). *Let $S \subseteq [p]$ be a set of $m$ indices. Consider the following linear regression model with zero vector as the true parameters:*

$$\boldsymbol{y}^{(k)} = \boldsymbol{X}_S^{(k)}\boldsymbol{0} + \boldsymbol{w}^{(k)}, \quad \text{for } k \in [K]$$

*where $\boldsymbol{y}^{(k)} = \boldsymbol{w}^{(k)} \in \mathbb{R}^n$, $\boldsymbol{X}_S^{(k)} \in \mathbb{R}^{n\times m}$ and $\boldsymbol{0} \in \mathbb{R}^m$. Assume that for each $k$, the row vectors of $\boldsymbol{X}^{(k)}$ are i.i.d. sampled from $\mathcal{N}(0, \Sigma^{(k)})$ and the noise is sampled from $\boldsymbol{w}^{(k)} \sim \mathcal{N}(0, \sigma_W^{(k)2}I_n)$. Denote $\rho := \max_{k\in[K]} \Sigma_{jj}^{(k)}$ and $\sigma_0 := \max_{k\in[K]} \sigma_W^{(k)}$. Consider the following $l_1/l_2$-regularized Lasso problem:*

$$\widehat{B} = \arg\min_{B\in\mathbb{R}^{m\times K}} \frac{1}{2n}\sum_{k=1}^K \|\boldsymbol{y}^{(k)} - \boldsymbol{X}_S^{(k)}\beta^{(k)}\|_2^2 + \lambda\|B\|_{l_1/l_2}, \tag{20}$$

*where $B = [\beta^{(1)}, \cdots, \beta^{(k)}]$. For any $\delta \in (0, 1)$, if*

$$t^* = \frac{\delta^2\lambda^2 n}{16\rho\sigma_0^2} > K,$$

*then with probability at least*

$$1 - m\exp\left( -t^*\left[ 1 - 2\sqrt{\frac{K}{t^*}} \right] \right) - K\exp\left( -\frac{4(1-\delta)}{\delta^2}n \right),$$

*$\widehat{B} = \boldsymbol{0}$ is an optimal solution to the problem in Eq. 20.*

The proof of this lemma is stated below, in which we simply use the notation $\boldsymbol{X}^{(k)}$ to replace $\boldsymbol{X}_S^{(k)}$.

**Lemma B.4.** *Suppose there exists a primal-dual pair $(\widehat{B}, \widehat{Z}) \in \mathbb{R}^{m\times K} \times \mathbb{R}^{m\times K}$ which satisfies the following conditions:*

$$\widehat{Z} \in \partial\|\widehat{B}\|_{l_1/l_2}, \tag{21a}$$

$$-\frac{1}{n}\boldsymbol{X}^{(k)\top}\left( \boldsymbol{y}^{(k)} - \boldsymbol{X}^{(k)}\widehat{\beta}^{(k)} \right) + \lambda\widehat{\boldsymbol{z}}^{(k)} = 0, \quad \forall k \in [K], \tag{21b}$$

$$\|\widehat{Z}\|_{l_\infty/l_2} < 1, \tag{21c}$$

*where $[\widehat{\beta}^{(1)}, \cdots, \widehat{\beta}^{(K)}]$ are the columns of $\widehat{B}$ and $[\widehat{\boldsymbol{z}}^{(1)}, \cdots, \widehat{\boldsymbol{z}}^{(K)}]$ are the columns of $\widehat{Z}$. Then $\boldsymbol{0}$ is the solution to the problem in Eq. 7 and it is the only solution.*

*Proof.* Straightforward by Lemma 1 in [19]. □

Therefore, to show $\mathbf{0} \in \arg\min_{B \in \mathbb{R}^{m \times K}} \frac{1}{2n} \sum_{k=1}^{K} \|\mathbf{y}^{(k)} - \mathbf{X}^{(k)}\beta^{(k)}\|_2^2 + \lambda\|B\|_{l_1/l_2}$, it is sufficient to show the existence of $(\widehat{B}, \widehat{Z})$ which satisfies the conditions in Eq. 21. We construct such a pair by the following definitions:

$$\widehat{B} := \mathbf{0}, \tag{22}$$

$$\widehat{\mathbf{z}}^{(k)} := \frac{1}{\lambda n} \mathbf{X}^{(k)\top} \mathbf{y}^{(k)}. \tag{23}$$

Clearly, they satisfy Eq. 21b. If we can show $\|\widehat{Z}\|_{l_\infty/l_2} < 1$, then Eq. 21 holds. Therefore, in this proof, the main goal is the analyze $\Pr\left[|\widehat{Z}\|_{l_\infty/l_2} < 1\right]$.

Denote the row vectors of $\widehat{Z}$ as $\widehat{Z}_j := [\widehat{\mathbf{z}}_j^{(1)}, \cdots, \widehat{\mathbf{z}}_j^{(K)}]$. Then $\|\widehat{Z}\|_{l_\infty/l_2} = \max_{j \in [m]} \|\widehat{Z}_j\|_2$. By definition in Eq. 23,

$$\widehat{\mathbf{z}}_j^{(k)} = \frac{1}{\lambda n} \mathbf{X}_j^{(k)\top} \mathbf{y}^{(k)} = \frac{1}{\lambda n} \mathbf{X}_j^{(k)\top} \mathbf{w}^{(k)}.$$

Since the linear combination of Gaussian distribution is still Gaussian, then given $\mathbf{w}^{(1:K)}$, the variable $\|\widehat{Z}_j\|_2^2$ is equivalent to a Chi-squared random variable in distribution:

$$\|\widehat{Z}_j\|_2^2 \mid \mathbf{w}^{(1:K)} = \frac{1}{\lambda^2 n^2} \sum_{k=1}^{K} \left(\mathbf{X}_j^{(k)\top} \mathbf{w}^{(k)}\right)^2 \mid \mathbf{w}^{(1:K)}$$

$$\overset{d.}{=} \frac{1}{\lambda^2 n^2} \sum_{k=1}^{K} \Sigma_{jj}^{(k)} \|\mathbf{w}^{(k)}\|_2^2 \xi_{jk}^2 \quad \text{where } \xi_{jk} \sim \mathcal{N}(0,1)$$

$$\leq \frac{1}{\lambda^2 n^2} \max_{k \in [K]} \Sigma_{jj}^{(k)} \max_{k \in [K]} \|\mathbf{w}^{(k)}\|_2^2 \sum_{k=1}^{K} \xi_{jk}^2$$

By Lemma H.2, for any $\delta > 0$,

$$\Pr\left[\max_{k \in [K]} \|\mathbf{w}^{(k)}\|_2^2 \geq \sigma_W^{(k)2} 2n(1+\delta)\right] \leq K \exp\left(-n(1+\delta)\left[1 - 2\sqrt{\frac{1}{1+\delta}}\right]\right).$$

Therefore, for all $\delta > 0$,

$$\Pr\left[\max_j \|\widehat{Z}_j\|_2 < 1\right] \geq \Pr\left[\max_{j \in [p]} \sum_{k=1}^{K} \xi_{jk}^2 < \frac{\lambda^2 n}{2(1+\delta)\rho\sigma_0^2}\right] \Pr\left[\max_k \|\mathbf{w}^{(k)}\|_2^2 < 2n(1+\delta)\right]$$

where

$$\rho := \max_{k \in [K]} \Sigma_{jj}^{(k)} \quad \text{and} \quad \sigma_0 := \max_{k \in [K]} \sigma_W^{(k)}.$$

Take $t^* = \frac{\lambda^2 n}{4(1+\delta)\rho\sigma_0^2}$, then if $t^* > K$, we have

$$\Pr\left[\max_{j \in [p]} \sum_{k=1}^{K} \xi_{jk}^2 < 2t^*\right] \geq 1 - p\exp\left(-t^*\left[1 - 2\sqrt{\frac{K}{t^*}}\right]\right).$$

Rewrite $\delta = \frac{4}{\delta'^2} - 1$ for some $\delta' \in (0,1)$ so that $1 + \delta = \frac{4}{\delta'^2} > 4$. If $\lambda$ is taken to be some value that satisfies the condition

$$t^* = \frac{\delta'^2 \lambda^2 n}{16\rho\sigma_0^2} > K,$$

then

$$\Pr\left[\max_j \|\widehat{Z}_j\|_2 < 1\right] \geq 1 - p\exp\left(-t^*\left[1 - 2\sqrt{\frac{K}{t^*}}\right]\right) - K\exp\left(-\frac{4(1-\delta')}{\delta'^2}n\right).$$

### B.3 Proof of error in F-norm

Denote the error vector as $\Delta_j^{(k)} := \widehat{g}_j^{(k)}(\hat{\pi}) - \widetilde{g}_j^{(k)}(\hat{\pi})$. Then

$$\sum_{k=1}^{K} \frac{1}{2n} \|\boldsymbol{X}^{(k)}\widehat{G}^{(k)} - \boldsymbol{X}^{(k)}\widetilde{G}^{(k)}(\hat{\pi})\|_F^2 = \sum_{k=1}^{K}\sum_{j=1}^{p} \frac{1}{2n} \|\boldsymbol{X}^{(k)}\Delta_j^k\|_2^2$$

By Theorem 7.3 in [14], with probability at least $1 - \exp(-\log p - \log K)$, it holds for all $k \in [K]$ and $j \in [p]$ that

$$\frac{1}{\sqrt{n}}\|\boldsymbol{X}^{(k)}\Delta_j^{(k)}\|_2 \geq \left(3/4\Lambda_{\min} - 3\sigma_{\max}\sqrt{\frac{\hat{d}\left(\log p + \log K\right)}{n}} - \sqrt{\frac{4(\log p + \log K)}{n}}\right)\|\Delta_j^{(k)}\|_2$$

$$\geq \left(3/4\Lambda_{\min} - 3\sigma_{\max}\sqrt{\frac{\hat{d}\left(\log p + \log K\right)}{n}} - c\right)\|\Delta_j^{(k)}\|_2$$

where $\hat{d} := \sup_{j,k}\|\Delta_j^{(k)}\|_0$. If the sample size $n$ satisfies the condition with a suitable constant $\kappa(\Lambda_{\min})$:

$$n \geq \kappa(\Lambda_{\min})\hat{d}\left(\log p + \log K\right),$$

then $\frac{1}{\sqrt{n}}\|\boldsymbol{X}^{(k)}\Delta_j^{(k)}\|_2 \geq \kappa'(\Lambda_{\min})\|\Delta_j^{(k)}\|_2$ for some constant $\kappa'(\Lambda_{\min})$. Therefore,

$$\frac{1}{K}\sum_{k=1}^{K}\sum_{j=1}^{p}\|\Delta_j^{(k)}\|_2^2 \leq \frac{2}{\kappa'(\Lambda_{\min})^2}\sum_{k=1}^{K}\frac{1}{2nK}\|\boldsymbol{X}^{(k)}\widehat{G}^{(k)} - \boldsymbol{X}^{(k)}\widetilde{G}^{(k)}(\hat{\pi})\|_F^2$$

$$\leq \frac{2}{\kappa'(\Lambda_{\min})^2}\left(\kappa(\sigma_{\max})\frac{p\log p}{nK} + cg_{\max}\frac{s_0\lambda}{\sqrt{K}}\right),$$

which implies

$$\frac{1}{K}\sum_{k=1}^{K}\sum_{j=1}^{p}\|\Delta_j^{(k)}\|_2^2 = \mathcal{O}\left(\frac{p\log p}{nK} + \frac{s_0\lambda}{\sqrt{K}}\right) = \mathcal{O}\left(\frac{s_0\lambda}{\sqrt{K}}\right).$$

The last equation holds for the case when $\lambda = \sqrt{\frac{p\log p}{nK}}$.

## C   Proof of Theorem 3.2: Support Recovery

We are interested in showing that the support union of $\widehat{G}^{(1:K)}$ is the same as that of $\widetilde{G}_0^{(1:K)}$. To prove this, we can equivalently show the support union of $\widehat{g}_j^{(1:K)}$ is the same as that of $\widetilde{g}_j^{(1:K)}$ for any $j \in [p]$.

Now we state the proof of Theorem 3.2.

*Proof.* Given a permutation $\pi_0 \in \Pi_0$, the DAG structure learning problem is equivalent to solving $p$ separate group Lasso problems, where for each $j$, the following $l_1/l_2$-penalized group Lasso is solved:

$$\widehat{g}_j^{(1:K)}|_{S_j(\pi_0)} = \underset{B \in \mathbb{R}^{|S_j(\pi_0)| \times K}}{\arg\min} \sum_{k=1}^{K} \frac{1}{2n}\|\boldsymbol{X}_j^{(k)} - \left(\boldsymbol{X}^{(k)}|_{S_j(\pi_0)}\right)\beta^{(k)}\|_2^2 + \lambda\|B\|_{l_1/l_2}, \tag{24}$$

where $S_j(\pi_0) := \{i : \pi_0(i) < \pi_0(j)\}$, and we denote the columns of $B$ as $B = \left[\beta^{(1)}, \cdots, \beta^{(K)}\right]$. The proof in this section is based on a uniform control over all $j$. For each $j$, the estimation in form of Eq. 24 is called a *multi-design multi-response* (or multivariate) regression problem, which has been studied in the last decade [18, 19]. Our support recovery analysis is based on techniques for

analyzing multivariate regression problem in existing literature, but careful adaptation is needed to simultaneously handle a set of $p$ problems where $p$ is the dimension of the problem.

More precisely, we present the analysis of multivariate regression problems in Appendix D, where the main results are summarized in Theorem D.1. Then the results in Theorem 3.2 can be obtained with direct computations by applying Theorem D.1 to $p$ separate problems defined by Eq. 24 and taking a union bound over all $j \in [p]$.

$\square$

## D Support Union Recovery for Multi-Design Multi-response Regression

The analysis in this section can be independent of the other content in this paper. We first introduce the multivariate regression setting and notations below.

**Problem setting and assumptions.** Consider the following $K$ linear regression models

$$\boldsymbol{y}^{(k)} = \boldsymbol{X}^{(k)}\beta^{*(k)} + \boldsymbol{w}^{(k)}, \quad \text{for } k = 1, \cdots, K,$$

where $\boldsymbol{y}^{(k)} \in \mathbb{R}^n$, $\boldsymbol{X}^{(k)} \in \mathbb{R}^{n \times p}$, $\beta^{*(k)} \in \mathbb{R}^p$, and $\boldsymbol{w}^{(k)} \in \mathbb{R}^n$. Assume that for each $k$, the row vectors of $\boldsymbol{X}^{(k)}$ are i.i.d. sampled from $\mathcal{N}(0, \Sigma^{(k)})$ and the noise is sampled from $\boldsymbol{w}^{(k)} \sim \mathcal{N}(0, \sigma^{(k)2}I_n)$. Denote $S$ as the support union of true parameters $\{\beta^{*(k)}\}_{k \in [K]}$, i.e., $S := \{j : \exists k \in [K] \, s.t., \beta_j^{*(k)} \neq 0\}$, and $s = |S|$ as its size. Note that the $s$ in this section has a different meaning from $s$ in other sections. Furthermore, for the true parameters, we denote $B^* = [\beta^{*(1)}, \cdots, \beta^{*(K)}]$ as the matrix whose columns are $\beta^{*(k)}$. Besides, we use $B_j^*$ to denote the $j$-th row of $B^*$.

**Assumptions and definitions.** Consider the following list of assumptions and definitions:

1. There exists $\gamma \in (0, 1]$ such that $\|A\|_\infty \leq 1 - \gamma$, where $A_{js} = \max_{k \in [K]} \left| \left( \Sigma_{S^c S}^{(k)} (\Sigma_{SS}^{(k)})^{-1} \right)_{js} \right|$ for $j \in S^c$ and $s \in S$.

2. There exist constants $0 < \Lambda_{\min} \leq \Lambda_{\max} < \infty$ such that all eigenvalues of $\Sigma_{SS}^{(k)}$ are in $[\Lambda_{\min}, \Lambda_{\max}]$ for all $k = 1, 2, \cdots, K$.

3. $\rho_u := \max_{j \in S^c, k \in [K]} \left( \Sigma_{S^c S^c | S}^{(k)} \right)_{jj}$

4. $\sigma_{\max} := \max_{k \in [K]} \sigma^{(k)}$

5. $b_{\min} := \min_{j \in S} \left\| B_j^* \right\|_2 / \sqrt{K}$.

With the above assumptions, we are ready to present the theorem.

**Theorem D.1.** *Assume the problem setting and assumptions in this section stated above. Consider the following $l_1/l_2$-regularized Lasso problem:*

$$\min_{B \in \mathbb{R}^{p \times K}} \frac{1}{2n} \sum_{k=1}^{K} \|\boldsymbol{y}^{(k)} - \boldsymbol{X}^{(k)}\beta^{(k)}\|_2^2 + \lambda\|B\|_{l_1/l_2}, \tag{25}$$

*where $B = [\beta^{(1)}, \cdots, \beta^{(k)}]$. If the following condition holds*

$$n \geq \kappa_6 s \log p,$$
$$K \leq c_0 \log p$$

$$\sqrt{\frac{8\sigma_{\max}^2 \log p}{\Lambda_{\min} n}} + \frac{2}{\Lambda_{\min}} \sqrt{\frac{sp \log p}{nK}} = o\left(b_{\min}^*\right),$$

*then Eq. 25 has a unique solution $\widehat{B}$, and that with probability at least*

$$1 - c_1 K \exp\left(-c_2(n-s)\right) - \exp\left(-c_3 \log p\right) - s \exp\left(-c_4 K \log p\right),$$

*the support union of $\widehat{B}$ is the same as $S$, and that $\left\| \widehat{B} - B^* \right\|_{l_\infty/l_2} / \sqrt{K} = o(b_{\min})$.*

*In this statement, $\kappa_6$ is a constant depending on $\gamma, \Lambda_{\min}, \rho_u, \sigma_{\max}$ and $c_i$ are universal constants.*

## D.1 Proof of Theorem D.1

The proof is based on a constructive procedure as specified by Lemma D.1, which characterizes an optimal primal-dual pair for which the primal solution $\widehat{B}$ correctly recovers the support set $S$.

**Lemma D.1.** *Define a pair* $\widehat{B} = [\widehat{\beta}^{(1)}, \cdots, \widehat{\beta}^{(K)}]$ *and* $\widehat{Z} = [\widehat{\boldsymbol{z}}^{(1)}, \cdots, \widehat{\boldsymbol{z}}^{(K)}]$ *as follows.*

$$\widehat{B}_{S^c} := 0 \tag{26a}$$

$$\widehat{B}_S := \underset{B_S \in \mathbb{R}^{s \times K}}{\arg\min} \frac{1}{2n} \sum_{k=1}^{K} \|\boldsymbol{y}^{(k)} - \boldsymbol{X}_S^{(k)} \beta_S^{(k)}\|_2^2 + \lambda \|B_S\|_{l_1/l_2} \tag{26b}$$

$$\widehat{\boldsymbol{z}}_S^{(k)} := -\lambda^{-1} \left( \widehat{\Sigma}_{SS}^{(k)} (\widehat{\beta}_S^{(k)} - \beta_S^{*(k)}) - \frac{1}{n} \boldsymbol{X}_S^{(k)\top} \boldsymbol{w}^{(k)} \right) \tag{26c}$$

$$\widehat{\boldsymbol{z}}_{S^c}^{(k)} := -\lambda^{-1} \left( \widehat{\Sigma}_{S^c S}^{(k)} (\widehat{\beta}_S^{(k)} - \beta_S^{*(k)}) - \frac{1}{n} \boldsymbol{X}_{S^c}^{(k)\top} \boldsymbol{w}^{(k)} \right) \tag{26d}$$

*The following statements hold true.*

(a) *If the matrix* $\widehat{Z}_{S^c} := [\widehat{\boldsymbol{z}}_{S^c}^{(1)}, \cdots, \widehat{\boldsymbol{z}}_{S^c}^{(K)}]$ *defined by Eq. 26d satisfies*

$$\|\widehat{Z}_{S^c}\|_{l_\infty/l_2} < 1, \tag{27}$$

*then* $(\widehat{B}, \widehat{Z})$ *is a primal-dual optimal solution to the* $l_1/l_2$-*regularized Lasso problem in Eq. 25. Furthermore, any optimal solution* $\widehat{B}$ *to Eq. 25 satisfies* $\widehat{B}_{S^c} = \boldsymbol{0}$.

(b) *Define a matrix* $U_S = [\boldsymbol{u}_S^{(1)}, \cdots, \boldsymbol{u}_S^{(K)}]$ *whose column vectors are*

$$\boldsymbol{u}_S^{(k)} := \widehat{\beta}_S^{(k)} - \beta_S^{*(k)} = (\widehat{\Sigma}_{SS}^{(k)})^{-1} \left( \frac{1}{n} X_S^{(k)\top} \boldsymbol{w}^{(k)} - \lambda \widehat{\boldsymbol{z}}_S^{(k)} \right).$$

*If the conditions in (a) are satisfied, and furthermore,* $U_S$ *satisfies*

$$\frac{\|U_S\|_{l_\infty/l_2}}{\sqrt{K}} \le \frac{1}{2} b_{\min}^*, \tag{28}$$

*then* $\widehat{B}$ *correctly recovers the union support* $S$. *That is,*

$$\left\{ i \in [p] : \exists k \in [K] \; s.t. \; \widehat{\beta}_i^{(k)} \ne 0 \right\} = S.$$

*Remark* D.1. Note that the matrix $\widehat{Z}_S := [\widehat{\boldsymbol{z}}_S^{(1)}, \cdots, \widehat{\boldsymbol{z}}_S^{(K)}]$ defined by Eq. 26c is a dual solution to the restricted optimization in Eq. 26b, and thus satisfies $\widehat{Z}_S \in \partial \|\widehat{B}_S\|_{l_1/l_2}$.

*Proof.* The proof of Lemma D.1 (a) is similar to Lemma 1 in [19] and Lemma 2 in [18]. The proof of Lemma D.1 (b) is straightforward from the condition in Eq. 28. By definition of $b_{\min}$, Eq. 28 implies $\|\widehat{\beta}_j^{(1:K)}\|_2 \ge \|\beta_j^{*(1:K)}\|_2 - \|\widehat{\beta}_j^{(k)} - \beta_j^{*(k)}\|_2 \ge \frac{1}{2}\sqrt{K} b_{\min}^* > 0$ for any $j \in S$. $\qquad\square$

Based on Lemma D.1, if we can show the primal-dual pair defined in its statement can satisfy both conditions in Eq. 27 and Eq. 28, then the support recovery guarantee is proved. We provide the analysis of these two conditions in Appendix D.1.1 and Appendix D.1.4 respectively.

Collecting the results in Appendix D.1.1 and Appendix D.1.4, we conclude that, if the following conditions are satisfied:

$$n \ge \kappa_6 s \log p, \quad K \le \frac{5}{64} \log p,$$

$$\sqrt{\frac{8\sigma_{\max}^2 \log p}{\Lambda_{\min} n}} + \frac{2}{\Lambda_{\min}} \sqrt{\frac{sp \log p}{nK}} = o\left(b_{\min}^*\right),$$

then with probability at least

$$1 - 3K \exp\left(-\frac{n}{2}\left(\frac{1}{4} - \sqrt{\frac{s}{n}}\right)_+^2\right) - K \exp\left(-(1+\delta)(n-s)\left[1 - 2\sqrt{\frac{1}{1+\delta}}\right]\right)$$

$$- \exp\left(-2\frac{3}{4}\log p\right) - s\exp\left(-2K\log p\left(1 - 2\sqrt{\frac{1}{2\log p}}\right)\right)$$

$$\geq 1 - c_1 K \exp\left(-c_2(n-s)\right) - \exp\left(-c_3\log p\right) - s\exp\left(-c_4 K \log p\right),$$

conditions in Eq. 27 and Eq. 28 are satisfied and therefore $\widehat{B}$ correctly recovers the union support $S$.

### D.1.1 No false recovery: $\|\widehat{Z}_{S^c}\|_{l_\infty/l_2} < 1$

Denote the row vectors of $\widehat{Z}_{S^c}$ as $\widehat{Z}_j := [\widehat{z}_j^{(1)}, \cdots, \widehat{z}_j^{(k)}]$. Then

$$\|\widehat{Z}_{S^c}\|_{l_\infty/l_2} = \max_{j\in S^c}\|\widehat{Z}_j\|_2.$$

Since

$$\widehat{Z}_j = \underbrace{\mathbb{E}[\widehat{Z}_j \mid \boldsymbol{X}_S^{(1:K)}]}_{T_{j1}} + \underbrace{\mathbb{E}[\widehat{Z}_j \mid \boldsymbol{X}_S^{(1:K)}, \boldsymbol{w}^{(1:K)}] - \mathbb{E}[\widehat{Z}_j \mid \boldsymbol{X}_S^{(1:K)}]}_{T_{j2}}$$
$$+ \underbrace{\widehat{Z}_j - \mathbb{E}[\widehat{Z}_j \mid \boldsymbol{X}_S^{(1:K)}, \boldsymbol{w}^{(1:K)}]}_{T_{j3}},$$

to prove $\|\widehat{Z}_{S^c}\|_{l_\infty/l_2} < 1$, we resort to bound $\max_{j\in S^c}\|T_{ja}\|_2$ for $a = 1, 2, 3$ separately. The analyses of $T_{j1}$ and $T_{j2}$ largely follow the arguments in [19] and [18], so details are omitted for brevity. We summarize the results of these two terms below, after which we present the detailed analysis for $T_{j3}$.

### D.1.2 Analysis of $T_{j1}$ and $T_{j2}$

$T_{j1}$: Following the same arguments as the derivations of Equation (28) in [19] and Equation (39) in [18], we have $\max_{j\in S^c}\|T_{j1}\|_2 \leq 1 - \gamma$.

$T_{j2}$: Following the same arguments as the derivations of Equation (32) in [19], we have that

$$\max_{j\in S^c}\|T_{j2}\|_2 \leq (1-\gamma)\|\widehat{Z}_S - Z_S^*\|_{l_\infty/l_2} + (1-\gamma)\mathbb{E}[\|\widehat{Z}_S - Z_S^*\|_{l_\infty/l_2} \mid \boldsymbol{X}_S^{(1:K)}],$$

where the rows of $Z_S^*$ are defined as $Z_i^* := B_i^*/\|B_i^*\|$ for $i \in S$. Define the matrix $\Delta \in \mathbb{R}^{s\times K}$ with rows $\Delta_i := (\widehat{B}_i - B_i^*)/\|B_i^*\|_2$. By Lemma H.3, if $\|\Delta\|_{l_\infty/l_2} < 1/2$, then it holds true that

$$\max_{j\in S^c}\|T_{j2}\|_2 \leq 4(1-\gamma)\left(\|\Delta\|_{l_\infty/l_2} + \mathbb{E}[\|\Delta\|_{l_\infty/l_2} \mid \boldsymbol{X}_S^{(1:K)}]\right).$$

We will show later in the analysis pf $U_S$ that $\|\Delta\|_{l_\infty/l_2}$ is of order $o(1)$ with high probability.

### D.1.3 Analysis of $T_{j3}$

Following the same arguments as the derivations of Equation (36) in [19] and Equation (42) [18], we have that for each $j \in S^c$,

$$\text{given } \left(\boldsymbol{X}_S^{(1:K)}, \boldsymbol{w}^{(1:K)}\right),$$
$$\widehat{z}_j^{(k)} - \mathbb{E}[\widehat{z}_j^{(k)} \mid \boldsymbol{X}_S^{(1:K)}, \boldsymbol{w}^{(1:K)}] \overset{d.}{=} \sigma_{jk}\xi_{jk},$$

where

$$\begin{cases} \xi_{jk} \sim \mathcal{N}(0,1), \\ \sigma_{jk}^2 := (\Sigma_{S^c S^c|S}^{(k)})_{jj}M_k \leq \rho_u M_k, \\ M_k := \frac{1}{n}\widehat{z}_S^{(k)\top}(\widehat{\Sigma}_{SS}^{(k)})^{-1}\widehat{z}_S^{(k)} - \frac{1}{n^2\lambda^2}\boldsymbol{w}^{(k)\top}(\Pi_S^{(k)} - I_n)\boldsymbol{w}^{(k)}. \end{cases}$$

Therefore,

$$\text{given } \left( \boldsymbol{X}_S^{(1:K)}, \boldsymbol{w}^{(1:K)} \right),$$

$$\max_{j \in S^c} \| \widehat{\boldsymbol{z}}_j^{(k)} - \mathbb{E}[\widehat{\boldsymbol{z}}_j^{(k)} \mid \boldsymbol{X}_S^{(1:K)}, \boldsymbol{w}^{(1:K)}] \|_2^2 \overset{d.}{=} \max_{j \in S^c} \sum_{k=1}^K \sigma_{jk}^2 \xi_{jk}^2$$

$$\leq \rho_u \max_{k \in [K]} |M_k| \max_{j \in S^c} \sum_{k=1}^K \xi_{jk}^2 \tag{29}$$

*(1) Bounding $\max_{k \in [K]} |M_k|$.*

Bound the term $\frac{1}{n} \widehat{\boldsymbol{z}}_S^{(k)\top} (\widehat{\Sigma}_{SS}^{(k)})^{-1}, \widehat{\boldsymbol{z}}_S^{(k)}$ is based on the following relations:

$$\max_{k \in [K]} \| \boldsymbol{z}_S^{*(k)} \|_2 \leq \sqrt{s} \quad \text{(by definition)},$$

$$\max_{k \in [K]} \| (\widehat{\Sigma}_{SS}^{(k)})^{-1} \|_2 \leq \frac{2}{\Lambda_{\min}} \quad \text{w.p.} \geq 1 - K \exp\left( -\frac{n}{2} \left( \frac{1}{4} - \sqrt{\frac{s}{n}} \right)_+^2 \right) \quad \text{(Lemma 10 in [18])}.$$

Therefore, with probability $\geq 1 - K \exp\left( -\frac{n}{2} \left( \frac{1}{4} - \sqrt{\frac{s}{n}} \right)_+^2 \right)$,

$$\frac{1}{n} \left| \widehat{\boldsymbol{z}}_S^{(k)\top} (\widehat{\Sigma}_{SS}^{(k)})^{-1} \widehat{\boldsymbol{z}}_S^{(k)} \right| \leq \frac{1}{n} \| (\widehat{\Sigma}_{SS}^{(k)})^{-1} \|_2 \| \widehat{\boldsymbol{z}}_S^{(k)} \|_2^2 \leq \frac{1}{n} \frac{2s}{\Lambda_{\min}},$$

For the second term in $\max_{k \in [K]} |M_k|$, note that

$$\boldsymbol{w}^{(k)\top} (I_n - \Pi_S^{(k)}) \boldsymbol{w}^{(k)} \overset{d.}{=} \sigma^{(k)2} \sum_{j=1}^{n-s} \zeta_{jk}^2 \quad \text{with } \zeta_{jk} \sim \mathcal{N}(0, 1).$$

By Lemma H.2, for any $\delta > 0$,

$$\Pr\left[ \max_{k \in [K]} \sum_{j=1}^{n-s} \zeta_{jk}^2 \geq 2(1+\delta)(n-s) \right] \leq K \exp\left( -(1+\delta)(n-s) \left[ 1 - 2\sqrt{\frac{1}{1+\delta}} \right] \right).$$

To summarize, with probability at least

$$1 - K \exp\left( -\frac{n}{2} \left( \frac{1}{4} - \sqrt{\frac{s}{n}} \right)_+^2 \right) - K \exp\left( -(1+\delta)(n-s) \left[ 1 - 2\sqrt{\frac{1}{1+\delta}} \right] \right),$$

it holds that

$$\max_{k \in [K]} |M_k| < \frac{1}{n} \frac{2s}{\Lambda_{\min}} + \frac{2\sigma_{\max}^2 (1+\delta)(n-s)}{n^2 \lambda^2}.$$

*(2) Bounding $\max_{j \in S^c} \sum_{k=1}^K \xi_{jk}^2$.*

Combining the bound on $\max_{k \in [K]} |M_k|$ with Eq. 29, it implies

$$\max_{j \in S^c} \| T_{j3} \|_2^2 \leq \rho_u \left( \frac{1}{n} \frac{2s}{\Lambda_{\min}} + \frac{2\sigma_{\max}^2 (1+\delta)(n-s)}{n^2 \lambda^2} \right) \max_{j \in S^c} \sum_{k=1}^K \xi_{jk}^2$$

$$\implies \left\{ \max_{j \in S^c} \| T_{j3} \|_2 < \gamma \right\} \subseteq \left\{ \max_{j \in S^c} \sum_{k=1}^K \xi_{jk}^2 < \frac{\gamma^2}{2\rho_u} \frac{\lambda^2 n}{\lambda^2 s / \Lambda_{\min} + \sigma_{\max}^2 (1+\delta) \frac{n-s}{n}} \right\}.$$

What's left is to bound the term $\sum_{k=1}^K \xi_{jk}^2$. Take $t^* = \frac{\gamma^2}{4\rho_u} \frac{\lambda^2 n}{\lambda^2 s / \Lambda_{\min} + \sigma_{\max}^2 (1+\delta) \frac{n-s}{n}}$. If $t^* > K$, by Lemma H.2 and a union bound over $j \in S^c$, we have that

$$\Pr[\max_{j \in S^c} \sum_{k=1}^K \xi_{jk}^2 \geq 2t^*] \leq (p-s) \exp\left( -t^* \left[ 1 - 2\sqrt{\frac{K}{t^*}} \right] \right).$$

*(3) Collecting all results.*

To conclude, for any $\delta > 0$, if the following conditions are satisfied:

$$\|\Delta\|_{l_\infty/l_2} < \frac{1}{2},$$

$$t^* = \frac{\gamma^2}{4\rho_u} \frac{\lambda^2 n}{\lambda^2 s/\Lambda_{\min} + \sigma_{\max}^2 (1+\delta)\frac{n-s}{n}} > K, \quad (30)$$

then for any $\delta > 0$, with probability at least

$$1 - K\exp\left(-\frac{n}{2}\left(\frac{1}{4} - \sqrt{\frac{s}{n}}\right)_+^2\right) - K\exp\left(-(1+\delta)(n-s)\left[1 - 2\sqrt{\frac{1}{1+\delta}}\right]\right)$$

$$- (p-s)\exp\left(-t^*\left[1 - 2\sqrt{\frac{K}{t^*}}\right]\right),$$

it holds that

$$\max_{j \in S^c} \|T_{j3}\|_2 < \gamma.$$

*(4) Condition in Eq. 30.*

If we assume that

$$n \geq Cs\log p$$

for some constant $C$. Then

$$t^* \geq \frac{\gamma^2}{4\rho_u} \frac{\lambda^2 n}{\lambda^2 n/(C\Lambda_{\min}\log p) + \sigma_{\max}^2(1+\delta)\frac{n-s}{n}} = \frac{\gamma^2}{4\rho_u} \frac{1}{(C\Lambda_{\min}\log p)^{-1} + \sigma_{\max}^2(1+\delta)\frac{n-s}{\lambda^2 n^2}}.$$

With the specified choice of parameter $\lambda = \sqrt{p\log p/n}$, it implies

$$t^* \geq \frac{\gamma^2}{4\rho_u} \frac{\log p}{(C\Lambda_{\min})^{-1} + \sigma_{\max}^2(1+\delta)\frac{n-s}{np}}.$$

Assume $C$ is chosen such that $C \geq \Lambda_{\min}^{-1}\left(\frac{\gamma^2}{20\rho_u} - \sigma_{\max}^2(1+\delta)\frac{n-s}{np}\right)^{-1}$ which can be easily satisfied since $\frac{n-s}{np} < 1$. Then it implies $t^* \geq 5\log p$. To satisfy the condition in Eq. 30, it is sufficient to assume $K \leq \frac{5}{64}\log p$, which implies $K \leq \frac{1}{64}t^*$. Furthermore, it implies $\exp\left(-t^*\left[1 - 2\sqrt{\frac{K}{t^*}}\right]\right) < \exp\left(-3\frac{3}{4}\log p\right)$.

To conclude, if $\|\Delta\|_{l_\infty/l_2} < \frac{1}{2}$ and that

$$n \geq \kappa_6 s\log p, \quad K \leq \frac{5}{64}\log p,$$

then $\max_{j \in S^c} \|T_{j3}\|_2 < \gamma$ holdes with probability at least

$$1 - K\exp\left(-\frac{n}{2}\left(\frac{1}{4} - \sqrt{\frac{s}{n}}\right)_+^2\right) - K\exp\left(-(1+\delta)(n-s)\left[1 - 2\sqrt{\frac{1}{1+\delta}}\right]\right)$$

$$- (p-s)\exp\left(-3\frac{3}{4}\log p\right).$$

### D.1.4 No exclusion: $\frac{\|U_S\|_{l_\infty/l_2}}{\sqrt{K}} \leq \frac{1}{2}b_{\min}^*$

Eq. 26c implies that

$$\widehat{\beta}_S^{(k)} - \beta_S^{*(k)} = \left(\widehat{\Sigma}_{SS}^{(k)}\right)^{-1}\left(\frac{1}{n}\boldsymbol{X}_S^{(k)\top}\boldsymbol{w}^{(k)} - \lambda\widehat{\boldsymbol{z}}_S^{(k)}\right).$$

Define $\overline{\boldsymbol{w}}^{(k)} := \frac{1}{\sqrt{n}}(\widehat{\Sigma}_{SS}^{(k)})^{-1/2}\boldsymbol{X}_S^{(k)\top}\boldsymbol{w}^{(k)} \stackrel{d.}{=} \sigma^{(k)}\xi_k$ with $\xi_k \sim \mathcal{N}(0, I_p)$. Then

$$\widehat{\beta}_S^{(k)} - \beta_S^{*(k)} \stackrel{d.}{=} \underbrace{(\widehat{\Sigma}_{SS}^{(k)})^{-1/2}\frac{\overline{\boldsymbol{w}}^{(k)}}{\sqrt{n}}}_{A^{(k)}} - \underbrace{(\widehat{\Sigma}_{SS}^{(k)})^{-1}\lambda\widehat{\boldsymbol{z}}_S^{(k)}}_{B^{(k)}}.$$

Denote the $i$-the entry in the vector $A^{(k)}$ as $A_i^{(k)}$. Then for a fixed $i \in S$, the entry $\{A_i^{(k)}\}_{k\in[K]}$ are independent. Its easy to see that

$$A_i^{(k)} \mid X^{(1:K)} \stackrel{d.}{=} \frac{\sigma^{(k)}}{\sqrt{n}}\sqrt{\left((\widehat{\Sigma}_{SS}^k)^{-1}\right)_{ii}}\xi_{ik} \quad \text{with } \xi_{ik} \sim \mathcal{N}(0,1) \text{ and } \mathrm{Cov}(\xi_{ik}, \xi_{ik'}) = 0.$$

Therefore,

$$\max_{i\in S}\sum_{k=1}^K A_i^{(k)2} \mid X^{(1:K)} \leq \frac{\sigma_{\max}^2}{n}\max_{k\in[K]}\left\|(\widehat{\Sigma}_{SS}^{(k)})^{-1}\right\|_2 \max_{i\in S}\sum_{k=1}^K \xi_{ik}^2.$$

Since we have

$$\Pr\left[\max_{k\in[K]}\left\|(\widehat{\Sigma}_{SS}^{(k)})^{-1}\right\|_2 \leq \frac{2}{\Lambda_{\min}}\right] \geq 1 - K\exp\left(-\frac{n}{2}\left(\frac{1}{4} - \sqrt{\frac{s}{n}}\right)_+^2\right) \quad \text{by Lemma 10 in [18]},$$

$$\Pr\left[\max_{i\in S}\sum_{k=1}^K \xi_{ik}^2 \leq 4K\log p\right] \geq 1 - s\exp\left(-2K\log p\left(1 - 2\sqrt{\frac{1}{2\log p}}\right)\right) \quad \text{by Lemma H.2},$$

then with probability at least $1 - K\exp\left(-\frac{n}{2}\left(\frac{1}{4} - \sqrt{\frac{s}{n}}\right)_+^2\right) - s\exp\left(-2K\log p\left(1 - 2\sqrt{\frac{1}{2\log p}}\right)\right)$, it holds that

$$\max_{i\in S}\sqrt{\sum_{k=1}^K A_i^{(k)2}} \leq \sqrt{\frac{8\sigma_{\max}^2 K\log p}{\Lambda_{\min}n}}.$$

Tuning now to the term $B^{(k)}$:

$$\max_{i\in S}\sqrt{\sum_{k=1}^K B_i^{(k)2}} = \lambda\max_{i\in S}\sqrt{\sum_{k=1}^K \left(\boldsymbol{e}_i^\top(\widehat{\Sigma}_{SS}^{(k)})^{-1}\widehat{\boldsymbol{z}}_S^{(k)}\right)^2}$$

$$\leq \lambda\max_{i\in S}\sqrt{\sum_{k=1}^K \|(\widehat{\Sigma}_{SS}^{(k)})^{-T}\boldsymbol{e}_i\|_2^2\|\widehat{\boldsymbol{z}}_S^{(k)}\|_2^2} \quad \text{by Cauchy-Schwarz inequality}$$

$$\leq \lambda\max_{i\in S}\max_{k\in[K]}\|(\widehat{\Sigma}_{SS}^{(k)})^{-T}\boldsymbol{e}_i\|_2\sqrt{\sum_{k=1}^K \|\widehat{\boldsymbol{z}}_S^{(k)}\|_2^2} \leq \lambda\max_{k\in[K]}\|(\widehat{\Sigma}_{SS}^{(k)})^{-1}\|_2\sqrt{s}.$$

The last inequality holds because $\|\widehat{Z}_S\|_{l_\infty/l_2} \leq 1$. Applying Lemma 10 in [18] to $\max_{k\in[K]}\|(\widehat{\Sigma}_{SS}^{(k)})^{-1}\|_2$ again, with probability at least $1 - K\exp\left(-\frac{n}{2}\left(\frac{1}{4} - \sqrt{\frac{s}{n}}\right)_+^2\right)$, it holds that

$$\max_{i\in S}\sqrt{\sum_{k=1}^K B_i^{(k)2}} \leq \frac{2\lambda\sqrt{s}}{\Lambda_{\min}}.$$

To conclude, with probability at least

$$1 - 2K\exp\left(-\frac{n}{2}\left(\frac{1}{4} - \sqrt{\frac{s}{n}}\right)_+^2\right) - s\exp\left(-2K\log p\left(1 - 2\sqrt{\frac{1}{2\log p}}\right)\right), \qquad (31)$$

it holds that (with specified $\lambda = \sqrt{\frac{p \log p}{n}}$)

$$\|\widehat{B}_S - B_S^*\|_{l_\infty/l_2} \le \sqrt{\frac{8\sigma_{\max}^2 K \log p}{\Lambda_{\min} n}} + \frac{2}{\Lambda_{\min}}\sqrt{\frac{sp \log p}{n}}$$

$$\implies \frac{\|U_S\|_{l_\infty/l_2}}{\sqrt{K}} \le \sqrt{\frac{8\sigma_{\max}^2 \log p}{\Lambda_{\min} n}} + \frac{2}{\Lambda_{\min}}\sqrt{\frac{sp \log p}{nK}}.$$

Therefore, if

$$\sqrt{\frac{8\sigma_{\max}^2 \log p}{\Lambda_{\min} n}} + \frac{2}{\Lambda_{\min}}\sqrt{\frac{sp \log p}{nK}} \le \frac{1}{2}b_{\min}^*,$$

then Eq. 28 is satisfied with probability specified in Eq. 31.

# E  Invariant Sets

We often need to take a union control over all permutations $\pi \in \mathbb{S}_p$ in the proofs. However, in these steps, we often care about the connection matrices $\widetilde{G}^{(1:K)}(\pi)$ instead of the permutation $\pi$ itself. Therefore, we want to see whether we can control a fewer number of events instead of enumerating over all $p!$ many permutations. Alternatively, we want to should that, given $\Sigma^{(1:K)}$, the number of elements in the set $\{\widetilde{G}^{(1:K)}(\pi) : \pi \in \mathbb{S}_p\}$ can be fewer than $p!$.

We start with the following definition which specifies the population-level quantity that we are interested in.

*Definition* E.1 (Population SEM). For any $S \subseteq [p] \setminus \{j\}$, let

$$g_j^{(k)}(S) := \underset{g \in \mathbb{R}^p, \text{supp}(g) \subseteq S}{\arg\min} \mathbb{E}[X_j^{(k)} - g^\top X^{(k)}]^2, \tag{32}$$

where $X^{(k)}$ is the random variable that follows $\mathcal{N}(0, \Sigma^{(k)})$.

$g_j^{(k)}(S)$ is called the SEM coefficients for variable $X_j$ regressed on the nodes in $S$ [15, 17]. It is a population-level quantity that depends on $\Sigma^{(k)}$, but not on the sample $\boldsymbol{X}^{(k)}$. In [15, 17], this quantity is used for a similar purpose on the single DAG estimation task. It is easy to verify that $g_j^{(k)}(S_j(\pi)) = \widetilde{g}_j^{(k)}(\pi)$. Lemma E.1 summarizes the key observations.

**Lemma E.1.** *Let $S_j(\pi) = \{i : \pi(i) < \pi(j)\}$ and $\widetilde{g}_j^{(k)}(\pi)$ is the $j$-th column of $\widetilde{G}^{(k)}(\pi)$. Then*

$$g_j^{(k)}(S) = \widetilde{g}_j^{(k)}(\pi)$$

*for any set $S$ such that*

$$supp\big(\widetilde{g}_j^{(k)}(\pi)\big) \subseteq S \subseteq S_j(\pi). \tag{33}$$

*Since the set of union parents $U_j(\pi) := \cup_{k \in [K]} supp\big(\widetilde{g}_j^{(k)}(\pi)\big)$ satisfies Eq. 33, it implies that*

$$g_j^{(k)}(U_j(\pi)) = \widetilde{g}_j^{(k)}(\pi), \quad \forall k \in [K]. \tag{34}$$

A direct consequence of this Lemma E.1 is that:

$$\Big|\{\widetilde{g}_j^{(1:K)}(\pi) : \pi \in \mathbb{S}_p\}\Big| = \Big|\{\widetilde{g}_j^{(1:K)}(U_j(\pi)) : \pi \in \mathbb{S}_p\}\Big| \le \Big|\{U_j(\pi) : \pi \in \mathbb{S}_p\}\Big|.$$

Recall that $d_j := \max_{\pi \in \mathbb{S}_p}|U_j(\pi)|$. Then there are at most $\sum_{0 \le m \le d_j}\binom{p}{m}$ many elements in this set. Note that

$$\sum_{0 \le m \le d_j}\binom{p}{m} \le \sum_{0 \le m \le d}\binom{p}{m} \le p^d.$$

The last inequality holds for all $p \ge d \ge 2$. Therefore,

$$\Big|\{\widetilde{g}_j^{(1:K)}(\pi) : \pi \in \mathbb{S}_p\}\Big| \le \Big|\{U_j(\pi) : \pi \in \mathbb{S}_p\}\Big| \le \sum_{0 \le m \le d_j}\binom{p}{m} \le p^d. \tag{35}$$

## F    Details of Continuous Formulation

We first prove that $\widehat{T} \in \mathcal{T}_p$:

**Lemma F.1.** *If $(\widehat{G}^{(1:K)}, \widehat{T})$ is a pair of optimal solution to Eq. 14, then $\widehat{T}$ is in the discrete space $\mathcal{T}_p$. Equivalently, if $\widehat{T}$ is an optimal solution, then there exists a permutation $\hat{\pi} \in \mathbb{S}_p$ such that*

$$\widehat{T}_{ij} = \begin{cases} 1 & \text{if } \hat{\pi}(i) < \hat{\pi}(j), \\ 0 & \text{otherwise}. \end{cases} \tag{36}$$

*Proof.* By the constraint $h(T) = 0$ in Eq. 15, the graph structure induced by the matrix $\widehat{T}$ must be acyclic. Therefore, $\widehat{T}$ represents a DAG and has an associated topological order (causal order). Denote this order by $\hat{\pi}$. What remains is to show Eq. 36 holds true.

Assume there exists an entry $(i', j')$ such that $\hat{\pi}(i') < \hat{\pi}(j')$ and $\widehat{T}_{i'j'} \neq 1$. We construct the following pair of $(T, G^{(1:K)})$:

$$T := \begin{cases} T_{i'j'} = 1 \\ T_{ij} = \widehat{T}_{ij} \text{ for } (i,j) \neq (i',j') \end{cases} \qquad G^{(k)} := \begin{cases} G_{i'j'}^{(k)} = \widehat{T}_{i'j'} \cdot \widehat{G}_{i'j'}^{(1:K)} \\ G_{ij}^{(k)} = \widehat{G}_{ij}^{(k)} \text{ for } (i,j) \neq (i',j') \end{cases} \qquad \forall k \in [K].$$

It is constructed by modifying the $(i', j')$-th entries in the solution pair $(\widehat{T}, \widehat{G}^{(1:K)})$. It is easy to see that: (i) this constructed pair $(T, G^{(1:K)})$ is a feasible solution to Eq. 14; and (ii) $(T, G^{(1:K)})$ achieves a smaller objective value than $(\widehat{T}, \widehat{G}^{(1:K)})$.

The reason for (ii) is that after the modification, the matrix $\overline{G}^{(1:K)}$ remains unchanged. That is, $\widehat{T} \circ \widehat{G}^{(k)} = T \circ G^{(k)}$. Therefore, the squared loss and the group-norm in the objective remain unchanged. However, the term $\|\mathbf{1}_{p \times p} - T\|_F^2$ has been reduced by setting $T_{i'j'} = 1$.

This makes a contradiction to the optimality of $(\widehat{T}, \widehat{G}^{(1:K)})$. Therefore, the assumption is not true and we conclude that:

$$\hat{\pi}(i) < \hat{\pi}(j) \Longrightarrow \widehat{T}_{ij} = 1.$$

Finally, since $\widehat{T}$ is consistent with $\hat{\pi}$, by definition, $\widehat{T}_{ij} = 0$ if $\hat{\pi}(i) \geq \hat{\pi}(j)$.    □

We now start to show the equivalence between the optimization in Eq. 7 and in Eq. 14.

Firstly, the solution search spaces are the same. We have shown that $\widehat{T} \in \mathcal{T}_p$. For each element in $\mathcal{T}_p$, we denote it by $\widehat{T}(\pi)$ based on its associated order $\pi$. Since $\widehat{T}(\pi)$ is a *dense* DAG with topological order $\pi$, it is easy to see the space $\{G \circ \widehat{T}(\pi) : G \in \mathbb{R}^{p \times p}\}$ includes all DAGs that are consistent with $\hat{\pi}$ and excludes any DAGs that are not. In other words, $\{G \circ \widehat{T}(\pi) : G \in \mathbb{R}^{p \times p}\} = \mathbb{D}(\pi)$. Therefore, the solution search spaces of these two optimization problems are equivalent.

Secondly, the optimization objectives are the same. Again, since $\widehat{T} \in \mathcal{T}_p$, the term $\rho \|\mathbf{1}_{p \times p} - \widehat{T}\|_F^2$ is a constant with a fixed value $\frac{\rho(p-1)p}{2}$. The remaining two terms in the objective are the same as the objective in Eq. 7.

Since both the solution search space and the optimization objectives are equivalent, these two optimizations are equivalent.

## G    Details of Synthetic Experiments in Sec 6.1

### G.1    Evaluation of structure prediction

We classify the positive predictions in three types:

- True Positive: predicted association exists in correct direction.
- Reverse: predicted association exists in opposite direction.

- False Positive: predicted association does not exist

Based on them, we use five metrics:

- False Discovery Rate (FDR): (reverse + false positive) / (true positive + reverse + false positive)

- True Positive Rate (TPR): (true positive) / (ground truth positive)

- False Positive Rate (FPR): (false positive) / (ground truth positive)

- Structure Hamming Distance (SHD): (false negative + reverse + false positive)

- Number of Non-Zero (NNZ): (true positive + reverse + false positive)

## G.2 A more complete result for Fig 3

We demonstrate our methods on synthetic data with $(p, s) \in \{(32, 40), (64, 96), (128, 224), (256, 512)\}$, $K \in \{1, 2, 4, 8, 16, 32\}$, $n \in \{10, 20, 40, 80, 160, 320\}$. For each $\{p, s, K, n\}$, we run experiments on 64 graphs. We report the full results in Fig.5.

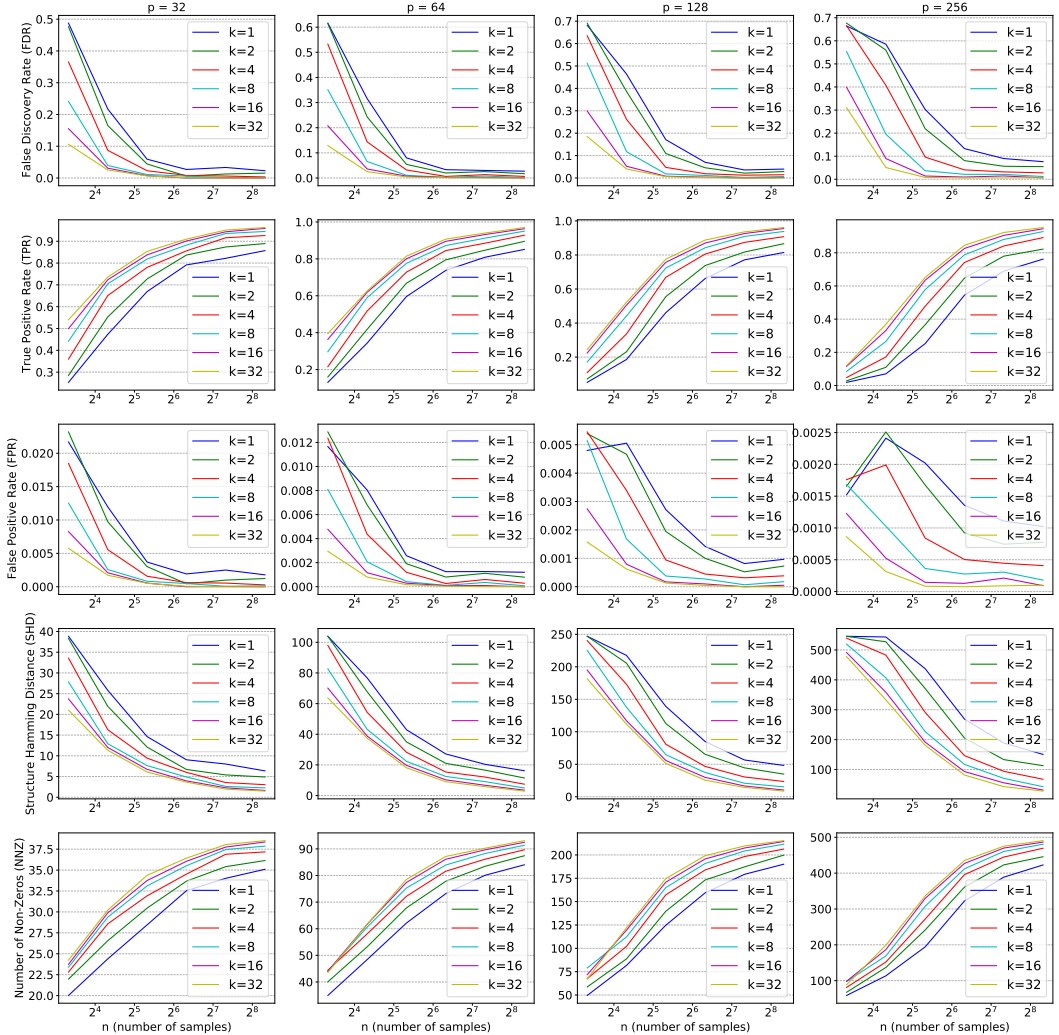

Figure 5: Full results of FDR, TPR, FPR, SHD, and NNZ.

### G.3  Computing resources

Since we need to run a large set of experiments spanning different values of $\{p, s, K, n\}$, the synthetic experiments are run on a CPU cluster containing 416 nodes. On each node, there are 24 CPUs (Xeon 6226 CPU @ 2.70GHz) with 192 GB memory. Each individual experiment is run on 4 CPUs. It takes about 10 hours to finish a complete set of experiments on about 400 CPUs.

# H   Useful Results In Existing Works

**Lemma H.1** (Laurent-Massart). *Let $a_1, \cdots, a_m$ be nonnegative, and set*

$$\|a\|_\infty = \sup_{i \in [m]} |a_i|, \quad \|a\|_2^2 = \sum_{i=1}^m a_i^2.$$

*For i.i.d $Z_i \sim \mathcal{N}(0, 1)$, the following inequalities hold for any positive $t$:*

$$\Pr\left[\sum_{i=1}^m a_i(Z_i^2 - 1) \geq 2\|a\|_2\sqrt{t} + 2\|a\|_\infty t\right] \leq e^{-t},$$

$$\Pr\left[\sum_{i=1}^m a_i(Z_i^2 - 1) \leq -2\|a\|_2 t\right] \leq e^{-t}.$$

**Lemma H.2.** *[18] Let $Z$ be a central Chi-squared distributed random variable with the degree $m$. Then for all $t > m$, we have*

$$\Pr[Z \geq 2t] \leq \exp\left(-t\left[1 - 2\sqrt{\frac{m}{t}}\right]\right).$$

**Lemma H.3.** *[18]   Consider the matrix $\Delta \in \mathbb{R}^{s \times K}$ with rows $\Delta_i := (\widehat{B}_i - B_i^*)/\|B_i^*\|_2$. If $\|\Delta\|_{l_\infty/l_2} < \frac{1}{2}$, then $\|\widehat{Z}_S - Z_S^*\|_{l_\infty/l_2} \leq 4\|\Delta\|_{l_\infty/l_2}$.*