# OpenReview forum: "Multi-task Learning of Order-Consistent Causal Graphs"
_NeurIPS.cc/2021/Conference — NeurIPS 2021 Poster_

### Official Review · Reviewer_x4HL · 2021-07-16

**Rating:** 6
**Confidence:** 4

**Summary:**

The paper considers structure discovery of multiple but related DAGs in the sense that they share a common causal order. Authors propose an l1/l2-regularized MLE for multiple linear Gaussian models and design a continuous optimization method to obtain the estimates. Sample complexity analysis are also provided regarding causal order and union support recovery, which is interesting and novel. Experiments also validate the theoretical finding and the effectiveness of joint estimation. The paper is generally well-written, though some details seem missing in the experiment part.

**Limitations And Societal Impact:**

I did not find any limitations and negative societal impact of the submission.

**Main Review:**

### Pros:

- A quantitative theoretic analysis of sample complexity regarding causal order and union support recovery, which is interesting and novel.
- An efficient algorithm to estimate the graph.
- Paper is generally very well-written.

### Cons:

Perhaps the most major issue to me is the setting of the problem.
-	Authors consider linear-Gaussian models where all the noise variances are identical, in particular, are set to be 1 among all datasets. The consideration for this choice is from identifiability. However, it is believed (at least I believe) that the condition of identical noise variances is quite restricted in practice, and it is not testable from observations. Consequently, the proposed theory and algorithm may have little use in practice.
-	Authors are encouraged to also consider non-identifiable case. In this case, it might be beneficial to reduce some indeterminacies of the edges in Markov equivalence class, since you have some different but related DAGs?
-	While it is claimed the analysis could easily extend to a more general case where the noise variances are different across datasets, it is not clear how they would affect the structure recovery result. In particular, if a dataset has a small very noise variance while another has a large one, then is the result affected?
-	What if the DAGs do not share the same causal order but can have a very few violations? Can the proposed algorithm return any guaranteed results?

Other issues in the experiment part:
-	The paper is well-written in other parts, but this section seem to omit some import details: 1) running time is not reported. 2) since you have multiple DAGs, it is not clear what DAGs are counted in calculating the metrics (FDR, SHD); are they the average? or simply the result regarding the causal order? 3) standard deviations are not provided.
-	Import baselines are missing: for linear Gaussian models with equal noise variances, a well known method is to check the marginal variances of the variables; the largest one would be the source node; for other variables, apply linear regression with the previous variable and obtain residuals; again check the marginal variances of these residuals and follow the same steps; one would then obtain a causal order. How does this procedure compare with the proposed method, when, e.g., in the second experiment that a single dataset can have a number of samples?
-	It would be also beneficial to consider some more settings: different graph structures and densities; different sample numbers across datasets.


**update after reading authors' response**

I appreciate authors' response which clearly extends the practical use of this work. I have decided to increase my score to 6 (tentatively), and I am also waiting for other reviewers' opinions on the pratical usage to see if it is possible to further increase the evaluation.

Two followed questions:
1. Even without identifiability of the unique graph, how about thinking about the MEC? i.e., how close the estimated graph could be to the MEC?
2. Please check the followed comment on the mentioned baseline. Sorry for any inconvience.



**Time Spent Reviewing:**

2.5

---

> ### Author Response · Authors · 2021-08-10
> **Response to the main concern - generalize the analysis to more practical setting**
>
> We will first address your major concern about the limitation of our setting, by explaining how our analysis can be extended to the following 3 more practical settings:
> 1. *Identifiable:* noise variances are identical within a single task, but can be different across tasks.
> 2. *Partially non-identifiable:* only $K'$ tasks satisfy the equal variance condition, but the other $K-K'$ are non-identifiable.
> 3. *Completely non-identifiable:* none of the $K$ tasks satisfies the equal variance condition.
>
> Our analysis can extend to the above settings 1 & 2 with **very few modifications** to the existing analysis. We could include them within the 9-page limit. For the 3rd setting, we are able to figure out how to analyze it. However, it will require a lot more spaces for stating the conditions and new results. We prefer to leave it to future work.
>
> Overall, we sincerely thank you for these constructive comments. It is interesting to see how non-identifiable DAGs can become identifiable under the multi-task learning setting. It helped us improve our paper!
>
> Details for each setting are presented below:
>
> ---
>
> ** **1. noise variances are identical within a single task, but can be different across tasks** **
>
> *Statement*:
>
> Let $\sigma_*^{(k)}{}^2$ be the variance of the $k$-th task. We only need to generalize the constant $\eta_w$ in Condition 3.4 to the following:
>
> $\frac{1}{pK}\sum_{k=1}^K \sum_{j=1}^p \left( \sigma_j^{(k)}(\pi)^2 - \sigma_*^{(k)}{}^2 \right)^2 > \frac{1}{\eta_w},\quad \forall \pi\notin\Pi_0$.
>
> Then the two main theorems hold true.
>
> *Explanation*:
> - The constant $\eta_w$ indicates how difficult it is to separate the true order from the other orders $\pi\notin\Pi_0$, by measuring the gap between true variance $\sigma_*^{(k)}$ and the variance $\sigma_j(\pi)$ with wrong order $\pi\notin\Pi_0$. If the gap is larger, separation is easier. Eventually, the order recovery mainly depends on this gap (i.e., easiness of separation). The noise variances can be different across datasets, as long as the averaged gap is larger than $1/\eta_w$ as defined in Condition 3.4.
> - The specific value of noise variance contributes to the maximal variance $\sigma_\text{max}^2$ which is bounded by the largest eigenvalue $\Lambda_\text{max}$ in the analysis.
> - After obtaining the order recovery guarantee, for the support recovery analysis, the original proof in the Appendix does not assume $\sigma_*^{(k)}=1$ and does not assume equal variance.
>
> *Proof*:
>
> It is virtually identical to the current proof, except that:
> - For Lemmas in Appendix, definitions of several constants need similar modification of replacing $1$ by $\sigma_*^{(k)}$, but all these constants are eventually bounded by $\Lambda_\text{max}$.
> - In many equations, the constant $1$ needs to be replaced by $\sigma_*^{(k)}$.
>
> ---
>
> ** **2. $K'$ tasks are identifiable, but $K-K'$ tasks are non-identifiable** **
>
> *Statement*:
>
> (i) Assume $K'$ models satisfy the equal variance condition (i.e., Condition 3.1). In this case, the true DAG $G_0^{(k)}$ is the **unique** minimum-trace DAG (line 83).
>
> (ii) Assume the other $K-K'$ models do not satisfy the equal variance condition, but for each model the true DAG $G_0^{(k)}$ is a minimum-trace DAG. These models are non-identifiable because the minimum-trace DAGs may not be unique without the equal variance condition.
>
> (iii) Condition 3.4 only needs to apply to the $K'$ identifiable models:
>
> $\frac{1}{pK'}\sum_{k=1}^{K'} \sum_{j=1}^p \left( \sigma_j^{(k)}(\pi)^2 - \sigma_*^{(k)}{}^2 \right)^2 > \frac{1}{\eta_w},\quad \forall \pi\notin\Pi_0$.
>
> Then the order recovery result in Theorem 3.1 holds true with the following modification to the sample complexity in Eq (6):
>
> $\theta(n,K,p,s) := \frac{p}{s}\sqrt{\frac{n}{p\log p}\frac{K'{}^2}{K}}>\kappa_1 \eta_w$.
>
> *Remark*:
> - Now the effective sample size indicated by $\theta$ is $n \cdot \frac{K'{}^2}{K}$.
> - If $K'=K$, it reverts back to the original result where the effective sample size is $nK$.
> - The support recovery analysis does not need modification.
>
> *Explanation*:
> - For a single non-identifiable task, it is impossible to distinguish the true DAG from other minimum-trace DAGs even with infinite samples.  All minimum-trace DAGs equally minimize the estimation objective when $n\rightarrow \infty$.
> - In the multi-task setting, we assume the $K$ models share the same topological order. Since **only one** topological order is optimized for **all** $K$ tasks, the identifiable $K'$ tasks can help to allow the correct order to minimize the joint objective. Therefore, the true DAG of a non-identifiable task can also be recovered.
> - Since the separation of the true order from the others mostly rely on the identifiable tasks, Condition 3.4 only needs to apply to the $K'$ tasks.
>
> *Brief proof*:
>
> The original proof is based on the bounds for two terms (I) and (II) in line 543 in Appendix. We split the summation in term (I) into two summations: $\sum_{k=1}^{K'}$ and $\sum_{k=K'}^K$. The former summation corresponds to identifiable models, which is still bounded by Lemma B.1. The latter is bounded by $\sigma_\text{max}^2 p(\sqrt{\frac{2(K-K')\log p }{n}} + \frac{2\log p}{n})$ with high probability, using Lemma H.1. Overall, we can bound term (I) with high probability as follows.
>
> $(I) \leq \underbrace{c_1 \frac{p\log p}{n} - c_2 \sum_{k=1}^{K'}\sum_{j=1}^p(1- \sigma_j^{(k)}(\hat{\pi})^2)^2}_{\text{from identifiable models}}$
>
> $ +\underbrace{ c_3 p(\sqrt{\frac{2(K-K')\log p }{n}} + \frac{2\log p}{n}) }_{\text{from non-identifiable models}}$
>
>
> Note that in the above bound, if one takes $K'=K$, the 3rd term becomes $0$ and the first two terms recover the original bound. The remaining derivation steps follow the same logic as the original proof steps.
>
> ---
>
> ** **3. all $K$ tasks are non-identifiable** **
>
> As indicated by setting 2, when all tasks are non-identifiable (i.e., $K'=0$), multi-task learning cannot help to recover the true order without further assumptions. An example is that all $K$ datasets are generated from the SAME model. In this case, no matter how many samples are available, there is no hope to distinguish the true DAG from other minimum-trace DAGs.
>
> However, if more conditions/assumptions can be imposed, it is possible to distinguish the true DAG under the multi-task learning. More specifically:
> - For each task, there is a set of minimum-trace DAGs. Let $\Pi^{(k)}$ be the set of their corresponding topological orders. Note that the true order $\pi_0$ is in the set $\Pi^{(k)}$ because the true DAG is assumed to be a minimum-trace DAG.
> - Apparently $\pi_0 \in \Pi^{(k)}$ for all $k$. To separate $\pi_0$ from other orders, we need to require $\pi_0$ to be **the only order** that appears in **every**  $\Pi^{(k)}$ for $k=1,\cdots,K$. That is, $\{\pi_0\} = \cap_{k=1}^K \Pi^{(k)}$, or $\{\Pi_0\} = \cap_{k=1}^K \Pi^{(k)}$ if there are more than one orders consistent with the true DAGs.
> - The final sample complexity will depend on $K'':= \max_{\pi\notin \Pi_0} \sum_{k=1}^K \mathbf{1}[\pi \in \Pi^{(k)}]$, which indicates how many sets $\Pi^{(k)}$ an **incorrect order** has appeared in. If the gap $K-K''$ is larger, then it is easier to distinguish the true order from others.
> - An interesting observation is that, when there are $K'$ identifiable tasks, then obviously $K-K''\geq K'$. Therefore, it will recover the result in setting 2.
>
> If required, we could also provide the detailed derivation steps on this openreview platform.

---

> > ### Author Response · Authors · 2021-08-10
> > **Response to other questions**
> >
> > Here we provide our responses to the second part of your comments.
> >
> > Firstly, regarding the setting *‘What if the DAGs do not share the same causal order but can have a very few violations?’*, we do not discuss it because our proposed joint estimator only estimates one causal order for all tasks. In case of violation, the consistency (recovery in the case of $n\rightarrow \infty$) does not hold. We think this setting is also very interesting and practical. However, we believe this will require the amount of work of another full paper.
> >
> > Now we respond to the remaining questions below.
> >
> > ---
> >
> > **Q1:** *The paper is well-written in other parts, but this section seem to omit some import details: 1) running time is not reported. 2) since you have multiple DAGs, it is not clear what DAGs are counted in calculating the metrics (FDR, SHD); are they the average? or simply the result regarding the causal order? 3) standard deviations are not provided.*
> >
> > *1) running time*
> >
> > We rerun the experiments on $K=256$ instances on cluster in single thread with Xeon 6226 CPU @ 2.70GHz, 8 Gb RAM. we report the average running time with its standard deviation:
> >
> >
> > Table 1: Running time for $p=64$ (in seconds).
> >
> > |        |   $n=10$  |  $n = 20$ |   $n=40$  |   $n=80$  |   $n=160$  |   $n=320$  |
> > |:------:|:---------:|:---------:|:---------:|:---------:|:----------:|:----------:|
> > |  K = 1 |  $64\pm0$ |  $75\pm1$ |  $77\pm1$ |  $76\pm1$ |  $87\pm1$  |  $104\pm2$ |
> > |  K = 2 |  $87\pm1$ |  $87\pm0$ |  $88\pm2$ |  $92\pm2$ |  $105\pm3$ |  $153\pm2$ |
> > |  K = 4 |  $89\pm1$ |  $90\pm1$ |  $98\pm2$ | $121\pm0$ |  $153\pm3$ | $266\pm14$ |
> > |  K = 8 |  $88\pm1$ |  $94\pm1$ | $121\pm0$ | $177\pm1$ |  $325\pm2$ |  $611\pm2$ |
> > | K = 16 |  $92\pm1$ | $117\pm0$ | $160\pm1$ | $304\pm2$ |  $534\pm1$ |  $936\pm1$ |
> > | K = 32 | $111\pm4$ | $175\pm8$ | $334\pm9$ | $532\pm3$ | $973\pm11$ | $1576\pm5$ |
> >
> > From this table, it is observed that the run time of this algorithm grows approximately (or slower than) linearly in $n$ and $K$.
> >
> > *2)* Yes, the reported results are averaged performance.
> >
> > *3) standard deviation*
> >
> > Thanks for the suggestion! Yes, we should provide the standard deviation to be more clear. The following is an example for the SHD with standard deviation. We will update the plots in the paper to include the error bar that shows the standard deviation for p=32,64,128,256 with FDR, TPR, FPR, and SHD.
> >
> > Table 2: SHD with standard deviation ($p=64$).
> >
> > |      |   $n=10$  | $n = 20$ |  $n=40$  |  $n=80$  |  $n=160$ |  $n=320$ |
> > |:----:|:---------:|:--------:|:--------:|:--------:|:--------:|:--------:|
> > | K=1  | $104\pm7$ | $77\pm8$ | $47\pm8$ | $29\pm6$ | $21\pm7$ | $19\pm8$ |
> > | K=2  | $103\pm7$ | $68\pm8$ | $38\pm7$ | $22\pm6$ | $16\pm5$ | $13\pm6$ |
> > | K=4  | $98\pm8$  | $54\pm7$ | $28\pm5$ | $17\pm4$ | $11\pm3$ | $9\pm4$  |
> > | K=8  | $85\pm7$  | $42\pm6$ | $22\pm4$ | $13\pm3$ | $8\pm3$  | $6\pm3$  |
> > | K=16 | $72\pm6$  | $37\pm5$ | $19\pm4$ | $11\pm3$ | $6\pm2$  | $4\pm2$  |
> > | K=32 | $66\pm5$  | $35\pm5$ | $17\pm3$ | $10\pm3$ | $6\pm2$  | $3\pm2$  |
> >
> > ---
> >
> > **Q2:** *Import baselines are missing: for linear Gaussian models with equal noise variances, a well known method is to check the marginal variances of the variables….. How does this procedure compare with the proposed method, when, e.g., in the second experiment that a single dataset can have a number of samples?*
> >
> > Thank you for giving us a detailed overview of a baseline we had overlooked. We implemented this baseline ourselves. When attempting to improve the baseline ROC curve for n = {50, 100}, we noticed that the strongest interactions identified by the baseline were always spurious: as we thresholded small values, the TPR always reached 0 before FDR ever dipped below 0.28. Table 3 demonstrates the set of results in the case of $n=100$.
> >
> > Table 3: SERGIO experiments with $n=100$.
> >
> > |                   | threshold |        FDR       |        TPR        |         FPR        |
> > |:-----------------:|:---------:|:----------------:|:-----------------:|:------------------:|
> > | Baseline          | 0.0       | $0.46\pm 1.3e-1$ | $0.81 \pm 1.8e-2$ | $0.022\pm 0.01$    |
> > | Baseline          | 0.01      | $0.32\pm 5.1e-2$ | $0.51 \pm 2.7e-2$ | $0.007 \pm 1.9e-3$ |
> > | Baseline          | 0.05      | $1.0\pm 0.0$     | $0.0 \pm 0.0$     | $0.005 \pm 1.4e-3$ |
> > | Baseline          | 0.1       | $1.0\pm 0.0$     | $0.0 \pm 0.0$     | $0.003 \pm 8.6e-4$ |
> > | Baseline          | 0.5       | $0.0\pm 0.0$     | $0.0 \pm 0.0$     | $0.0 \pm 0.0$      |
> > | Our method (K=1)  | 0.8       | $0.18\pm 3.8e-3$ | $0.30 \pm 1.2e-3$ | $0.001 \pm 7.3e−7$ |
> > | Our method (K=10) | 0.8       | $0.07\pm 1.2e-3$ | $0.42 \pm 1.2e-3$ | $0.001 \pm 2.8e−7$ |
> > | Our method (K=10) | 0.8  | $0.07\pm 1.2e-3$ | $0.42 \pm 1.2e-3$ | $0.001 \pm 2.8e−7$ |
> >
> > In all cases, we observed that the baseline has a high FDR in the high-dimensional scenario n<p.
> >
> > Furthermore, as suggested by Reviewer T7T2, we include a comparison to a joint estimation baseline. Since we are mainly looking at the multi-task setting, this comparison to a joint estimation algorithm might be more meaningful. We would like to kindly refer you to our response to Reviewer T7T2 (Question 3).

---

> > > ### Comment · Reviewer_x4HL · 2021-08-13
> > > **Correction of the baseline**
> > >
> > > Sorry, just noticed the prevous comment needs a correction:
> > >
> > > *Import baselines are missing: for linear Gaussian models with equal noise variances, a well known method is to check the marginal variances of the variables; the **largest** one would be the source node; for other variables, apply linear regression with the previous variable and obtain residuals; again check the marginal variances of these residuals and follow the same steps; one would then obtain a causal order. How does this procedure compare with the proposed method, when, e.g., in the second experiment that a single dataset can have a number of samples?*
> > >
> > > **largest** should be the **smallest** one.  Please let me know if your implementation chooses the largest one or smallest one. Sorry again.

---

> > > > ### Author Response · Authors · 2021-08-18
> > > > **Thank you for the feedback and follow-up questions!**
> > > >
> > > > Thank you for the feedback on our response which is really appreciated!
> > > >
> > > > **Q1:** *Even without identifiability of the unique graph, how about thinking about the MEC? i.e., how close the estimated graph could be to the MEC?*
> > > >
> > > > We also thank you for the follow-up question regarding MEC.
> > > > - In the non-identifiable case, if one is interested in identifying the Markov equivalent class, then the $\ell_0$ penalized estimator is a more suitable choice than the $\ell_1$ penalized estimator we used in this paper (when $K=1$, our $\ell_2$/$\ell_1$ group-norm is $\ell_1$ norm). The reason is that $\ell_0$ penalization leads to invariant scores over Markov equivalent DAGs but $\ell_1$ penalization does not.
> > > > - For the single task scenario, [20] has provided the theoretical guarantee for recovering minimum-edge I-MAPs (which is the MEC if the “faithfulness” condition holds) using $\ell_0$ penalized estimator.
> > > > - To estimate MEC in case of non-identifiability, one might adapt our analysis framework to extend the result in [20] to the multi-task scenario. The major difference from our current analysis is the $\ell_2$/$\ell_1$ group-norm may need to be modified to the $\ell_2$/$\ell_0$ group-norm and the corresponding analysis needs to change.
> > > > - Furthermore, the major reason why we do not consider $\ell_0$ based estimator is the computational complexity of the optimization, which isn’t practical especially for joint estimation.
> > > >
> > > > **Q2:** *Correction of the baseline*
> > > >
> > > > Yes, we indeed follow the description to select the node with *largest* marginal variance. We have now modified the code to choose the *smallest* one.
> > > > - However, it still works very badly on the SERGIO dataset. The reason might be that SERGIO generates data from a more realistic model which is not exactly linear SEM.
> > > > - Therefore, we apply this baseline to the synthetic dataset (Sec 6.1) generated from linear SEMs. Results are shown in the following table.
> > > > - In this table, “simple baseline” is the one that you described. Baseline [27] is a joint estimation algorithm. In the case of $K=1$, baseline [27] is essentially the popular GES algorithm for single DAG recovery.
> > > > - It can be observed that "simple baseline" performs badly in the case of $n<p$. However, in the large-sample setting ($n=320$), it performs better than baseline [27] and ours when $K=1$, but it is still worse than our joint estimation results.
> > > >
> > > > Table: SHD with standard deviation ($p=64$).
> > > >
> > > > |      |   $n=10$  | $n = 20$ |  $n=40$  |  $n=80$  |  $n=160$ |  $n=320$ |
> > > > |:----:|:---------:|:--------:|:--------:|:--------:|:--------:|:--------:|
> > > > | K=1 (simple baseline) | $119\pm8$ |  $106\pm10$ |  $76\pm10$  | $44\pm7$  | $25\pm6$ | $13\pm4$  |
> > > > | K=1 (baseline [27]) | $100\pm11$ | $53\pm11$ | $28\pm11$ | $18\pm9$ | $17\pm10$ | $18\pm9$  |
> > > > | K=1 (ours) | $104\pm7$ | $77\pm8$ | $47\pm8$ | $29\pm6$ | $21\pm7$ | $19\pm8$ |
> > > > | K=8 (baseline [27])  | $82\pm10$  | $42\pm10$ | $29\pm7$  | $22\pm6$ | $28\pm7$  | $27\pm8$  |
> > > > | K=8 (ours)  | $85\pm7$  | $42\pm6$ | $22\pm4$ | $13\pm3$ | $8\pm3$  | $6\pm3$  |

---

### Official Review · Reviewer_T7T2 · 2021-07-17

**Rating:** 6
**Confidence:** 4

**Summary:**

The paper proposes a multi-task DAG learning under equal variance and Gaussian assumptions. Consistency results are provided and new algorithm is proposed to solve the formulation. Empirically, results are shown to be consistent with the theoretical analysis.

**Limitations And Societal Impact:**

Did not find such a section.

**Main Review:**

Strength:

Recovery main results are serious work and appreciated.

Formulation of objective is novel

Weakness:

Strong assumptions on equal variance and Gaussian, although I understand the necessity for the theoretical analysis.

Empirical evaluation is very weak.


Comments:
- L51: it is not clear on what cardinality p is.
- The model and method only work with Gaussian data, so the current title and introduction is somewhat misleading. Authors should make it clear that the model under consideration is only Gaussian DAG.
- Equal variance and Gaussian assumptions: they are pretty strong assumption in the context of DAG identification and learning. Such a restricted space may limit its practical values.
- causal order recovery theoretical analysis is thorough (although I did not check the proof details)
- Typical Gaussian network structures are learned via precision matrices. It seems possible that these precision matrices can help learning the DAG, e.g., serve as an skeleton of the DAG learning. Have authors considered this aspect?
- "the optimal solution must be an element in the discrete space Tp": would be good to have a formal statement on this.
- Algorithm 1 uses a min-max formulation and solved slightly different from original NOTEAR. Can authors discuss the advantages of this method?
- There seem no baseline methods for comparison.  This is a major concern. Why are [27][28] not compared?
- Gene regulatory network: it seems K = 1, n = 1000 has a better TPR: 0.47 and better SHD.  The results are therefore not very convincing for the proposed approach. What about cases with K > 1 and n = 1k?

**Time Spent Reviewing:**

3.5

---

> ### Author Response · Authors · 2021-08-11
> **Response to Reviewer T7T2 (Part 1)**
>
> We thank the reviewer for the very careful reading and constructive comments! We respond to every raised question/comment below. Hope they can address your concerns.
>
> ---
>
> **Q1:** *Equal variance and Gaussian assumptions: they are pretty strong assumption in the context of DAG identification and learning. Such a restricted space may limit its practical values.*
>
> *(1) Equal variance condition:*
>
> This is also a major concern of Reviewer x4HL. We would like to kindly refer you to our response to Reviewer x4HL on this question.
>
> To summarize, we showed how our analysis/result can extend to the following two settings with **very few modifications** to the current paper:
>
> - Setting 1: We assume equal variance within a single task, but the variances can be different across different tasks. Note that in this case, each task is still identifiable.
>
> - Setting 2: We only impose the equal variance condition on $K’$ tasks, but allow the other $K-K’$ tasks not to satisfy this condition. Note that in this case, $K’$ tasks are identifiable, and $K-K’$ tasks are non-identifiable.
>
> Furthermore, we also describe how we can handle the more difficult setting where all $K$ tasks are non-identifiable. However, this case will require a lot more space to state, so we plan to mention it in a Discussion section and leave it to future work.
>
> Finally, thank you for raising this question. The extension that shows how non-identifiable DAGs can become identifiable under the multi-task learning setting is indeed interesting. It helped us improve our paper!
>
> *(2) Gaussian assumption:*
>
> It is natural to inquire how the results extend to non-Gaussian models.
>
> However, even for the single task DAG recovery, the non-Gaussian case hasn’t been well-understood yet. As mentioned in [21],  ‘the difficulty is that the **population-level structure** of such structural equations is not well understood in the non-Gaussian setting ... For example, for different permutations the errors $\varepsilon_j(\pi)$ are no longer guaranteed to be independent of the parents $X_{S_j}(\pi)$ if the joint distribution is not Gaussian.’
>
> In addition, the multi-task setting requires better characterization of the distribution properties such as tail behavior. To analyze non-Gaussian noise, it requires some fundamental statistical techniques to give tight concentration bounds as those used in our paper for Chi-squared distribution.
>
> It is an interesting future direction to consider extensions to non-Gaussian models. It may require applying techniques used in some existing works on non-Gaussian undirected graphs such as [A][B].
>
> Finally, as suggested, we will edit the paper to emphasize ‘Gaussian’ DAG to avoid confusion.
>
> [A] Liu, Han, John Lafferty, and Larry Wasserman. "The nonparanormal: Semiparametric estimation of high dimensional undirected graphs." Journal of Machine Learning Research 10.10 (2009).
>
> [B] Loh, Po-Ling, and Martin J. Wainwright. "Structure estimation for discrete graphical models: Generalized covariance matrices and their inverses." The Annals of Statistics (2013): 3022-3049.]
>
> ---
>
> **Q2.** *Typical Gaussian network structures are learned via precision matrices. It seems possible that these precision matrices can help learning the DAG, e.g., serve as an skeleton of the DAG learning. Have authors considered this aspect?*
>
> Yes! We indeed have thought about using a jointly estimated skeleton as the constrained support set for DAG recovery! The analysis can go through the following steps:
> 1. Provide an error bound for the joint estimation of support union, by using some known results in papers such as [C,D,E]. This support union estimation is usually based on the precision matrix estimation as you mentioned. The estimation should be a skeleton that contains the support union, and its size should be << $p^2$.
> 2. Assume that such a skeleton is given, then the DAG recovery analysis can go through another statistical framework, based on the analysis of uniform convergence of empirical norm. The required basic inequalities can be found in [F]. A recent paper [G] has used this technique to analyze single-task DAG recovery. Extension to multi-task setting isn’t too difficult.
>
> Overall, if a skeleton is known, the search space is largely reduced so a simpler analysis framework (step 2 mentioned above) can be applied. However, without knowing the skeleton, step 2 will not give a tight bound.
>
> We do not extend our paper to include this case because (1) there is a 9-page limit and (2) this will bring in a different analysis framework that’s not well connected with the current framework.
>
> [C] N. Meinshausen and P. Bühlmann. High-dimensional graphs and variable selection with the Lasso. Annals of Statistics, 34(3):1436–1462, 2006.
> [D] Honorio, Jean, and Dimitris Samaras. "Multi-task learning of Gaussian graphical models." ICML. 2010.
> [E] Zhang, Qian, Yilin Zheng, and Jean Honorio. "Meta Learning for Support Recovery in High-dimensional Precision Matrix Estimation." International Conference on Machine Learning. PMLR, 2021.
>
> [F] van de Geer, Sara. "On the uniform convergence of empirical norms and inner products, with application to causal inference." Electronic Journal of Statistics 8.1 (2014): 543-574.
> [G] Aragam, Bryon, Arash Amini, and Qing Zhou. "Globally optimal score-based learning of directed acyclic graphs in high-dimensions." Advances in Neural Information Processing Systems 32 (2019): 4450-4462.
>
> ---
>
> **Q3:** *There seem no baseline methods for comparison. This is a major concern. Why are [27][28] not compared?*
>
> - For [28], we found the released implementation. However, it performs badly on our datasets. Therefore, we read the paper carefully again and noticed that it assumes the knowledge of topological order. As stated on Page 4, ‘In this paper, we assume that a parent ordering of variables is known in which no edges exist from larger vertices to smaller vertices.’ Therefore, we believe this approach isn’t comparable. We will mention its assumption in our related work section to make it clear.
>
> - For [27], the implementation isn’t publicly available. We followed the algorithm box presented in the paper to implement their method. We select the parameter $\lambda_1^2 = 3 \frac{\log p}{n}$ as suggested by [27]. We picked $\lambda_2^2 = \sqrt{\frac{\log p}{8n}}$ by cross validation as suggested by [27]. We report the results for $p=64$ below.
>
> Table 1: Runtime of [27] in seconds
>
> |  | $n=10$ | $n = 20$ | $n=40$ | $n=80$ | $n=160$ | $n=320$ |
> |---|---|---|---|---|---|---|
> | k=1 | $558\pm28$ | $827\pm19$ | $824\pm17$ | $829\pm15$ | $824\pm15$ | $844\pm19$ |
> | k=2 | $909\pm22$ | $898\pm18$ | $894\pm15$ | $907\pm17$ | $887\pm17$ | $917\pm20$ |
> | k=4 | $1023\pm13$ | $1004\pm15$ | $995\pm15$ | $1024\pm23$ | $990\pm17$ | $1031\pm18$ |
> | k=8 | $1232\pm41$ | $1190\pm18$ | $1198\pm18$ | $1227\pm23$ | $1206\pm50$ | $1243\pm20$ |
> | k=16 | $1626\pm27$ | $1580\pm19$ | $1597\pm18$ | $1743\pm109$ | $1591\pm17$ | $1690\pm28$ |
> | k=32 | $2452\pm56$ | $2358\pm43$ | $2402\pm34$ | $2457\pm46$ | $2381\pm18$ | $2546\pm40$ |
> |  |  |  |  |  |  |  |
>
> Table 2: Performance of [27] in SHD ($p=64$)
>
> |      | $n=10$     | $n = 20$  | $n=40$    | $n=80$   | $n=160$   | $n=320$   |
> |------|------------|-----------|-----------|----------|-----------|-----------|
> | k=1  | $100\pm11$ | $53\pm11$ | $28\pm11$ | $18\pm9$ | $17\pm10$ | $18\pm9$  |
> | k=2  | $99\pm10$  | $51\pm10$ | $29\pm10$ | $20\pm9$ | $21\pm10$ | $21\pm10$ |
> | k=4  | $92\pm10$  | $47\pm10$ | $31\pm9$  | $22\pm7$ | $26\pm8$  | $25\pm9$  |
> | k=8  | $82\pm10$  | $42\pm10$ | $29\pm7$  | $22\pm6$ | $28\pm7$  | $27\pm8$  |
> | k=16 | $65\pm9$   | $37\pm8$  | $30\pm6$  | $20\pm5$ | $28\pm6$  | $25\pm6$  |
> | k=32 | $57\pm9$   | $36\pm6$  | $27\pm6$  | $19\pm5$ | $25\pm5$  | $26\pm6$  |
> | | | | | | | |
>
> We would like to kindly refer the reviewer to Table 1 & 2 in our response to Reviewer x4HL. There we report the run-time and SHD of our algorithm under exactly the same setting. It can be observed that:
> - These two algorithms have similar performance when $K=1$. However, when $K$ increases, our algorithm returns consistently better structures in terms of SHD.
> - In addition, our algorithm is much faster, because [27] is based on greedy equivalence search (GES).
> - We don’t have enough time to finish evaluating [27] on p=128, 256. For the instances we have solved, (p, k, n) = (128, 1, 10) requires 8912 seconds on average , and (p, k, n) = (128, 32, 320) requires 25361 seconds on average.

---

> > ### Author Response · Authors · 2021-08-11
> > **Response to Reviewer T7T2 (Part 2)**
> >
> > Here we provided our responses to the second part of Reviewer T7T2’s comments.
> >
> > ---
> >
> > **Q4:** *Gene regulatory network: it seems K = 1, n = 1000 has a better TPR: 0.47 and better SHD. The results are therefore not very convincing for the proposed approach.*
> >
> > We would like to claim that the better performance of (K=1,n=1000) should not make our approach unconvincing because we **do not** expect (K=10, n=100) to outperform (K=1, n=1000), as explained below.
> >
> > - In the case of (K=10, n=100), each task has only 100 samples. Although it can leverage 900 samples from other (similar) tasks, these 900 samples can be 'noisier’ than the 100 samples due to certain variations among the tasks. However, in the case of (K=1, n=1000), all 1000 samples come from the same task, so they are 'clean’ samples for this task. Therefore, it is expected to perform better than (K=10, n=100).
> >
> > - In fact, one should expect the performance of (K=1, n=1000) to be the upper bound of the performance of (K=10, n=100), and this upper bound can be reached when all K datasets are sampled from the same model. Therefore, our comparison between these two scenarios aims at showing that (K=10, n=100) is achieving a **comparable performance** as (K=1, n=1000), which in turns shows the proposed approach is able to **effectively leverage** samples from other similar tasks.
> >
> > - The set of results for (K=10, n=100) and (K=1, n=100) in Table 1 in the paper demonstrates a direct comparison between joint estimation and separate estimation to explicitly shows the **improvement** of multi-task learning. We will include the additional comparison between (K=20, n=50) and (K=1, n=50) as shown in the following table to provide more support to the effectiveness of our approach under the small-sample scenario (i.e., n<p). The dimension of this problem is p=100. For the large sample scenario n=1000, increasing the task numbers does not offer improvements as obvious as the small sample cases.
> >
> > | K | n | FDR | TPR | FPR | SHD |  |
> > |---|---|---|---|---|---|---|
> > | 1 | 50 | 0.11 $\pm$ 7e-2 | 0.19 $\pm$ 2e-2 | 0.0007 $\pm$ 4.7e-5 | 114.4 $\pm$ 3.9 |  |
> > | 20 | 50 | 0.1 $\pm$ 7e-2 | 0.38 $\pm$ 2e-2 | 0.001 $\pm$ 1e-3 | 90.54 $\pm$ 3.43 |  |
> > |  |  |  |  |  |  |  |
> > | 1 | 100 | 0.18 $\pm$ 3.8e-3 | 0.30 $\pm$ 1.2e−3 | 0.001 $\pm$ 7.3e−7 | 104.61 $\pm$ 3.2 |  |
> > | 10 | 100 | 0.07 $\pm$ 1.2e−3 | 0.42 $\pm$ 5.1e−4 | 0.001 $\pm$ 2.8e−7 | 84.0 $\pm$ 1.5 |  |
> > |  |  |  |  |  |  |  |
> > | 1 | 1000 | 0.09 $\pm$ 2.9e-3 | 0.50 $\pm$ 1.4e-3 | 0.002 $\pm$ 9.2e-7 | 76.4 $\pm$ 5.9 |  |
> > |  |  |  |  |  |  |  |
> >
> > To conclude, the goal of our approach is to effectively share information between many sample-limited tasks to improve DAG recovery, which we believe is a novel contribution we have achieved.
> >
> > ---
> >
> > **Q5** *Algorithm 1 uses a min-max formulation and solved slightly different from original NOTEAR. Can authors discuss the advantages of this method?*
> >
> > The original NOTEARS is also based on a min-max formulation. The main difference is NOTEARS solves the minimization by proximal quasi-Newton (PQN) method, which is a second-order method. The multi-task setting in case of high-dimension cannot afford the computational cost of such a method, because $K$ tasks need to be jointly optimized. We used a gradient based proximal method which is a first-order method. It’s main advantage is efficiency.
> >
> > ---
> >
> > **Q6** *clarity*
> >
> > Thanks for pointing out some confusing terminology in the paper. We will improve the clarity of the paper regarding the following two comments:
> >
> > - *"L51: it is not clear on what cardinality p is."* $p$ is the dimension of the problem. In L51, the cardinality is $p!$ ($p$ factorial), that equals to $p\cdot(p-1)\cdots 1$.
> >
> > - *"the optimal solution must be an element in the discrete space Tp": would be good to have a formal statement on this."* We will include a Lemma (either in the main text or Appendix, depending on the space) to formally state this observation.

---

### Official Review · Reviewer_Masb · 2021-07-19

**Rating:** 7
**Confidence:** 3

**Summary:**

Authors study the problem of learning the causal ordering and union support of DAGs from K linear SEMs. Authors propose a regularized joint estimator and show that this joint estimation enjoys a better sample complexity than single separate estimations. Moreover, guarantees for causal order and union support recovery are provided. For experiments, authors propose a continuous optimization problem that outputs the same causal order and union support than the joint estimator.

**Limitations And Societal Impact:**

I think the limitations can be stated more explicitly in a separate section or paragraph.

**Main Review:**

Here I present my review along the axes from the NeurIPS reviewing guidelines:

**Originality**

- Are the tasks or methods new?
  - Yes. Authors study the problem of multitask learning of causal ordering which has been previously unexplored to the best of my knowledge.

- Is the work a novel combination of well-known techniques?
  - It is motivated by existing methods for multitask learning in the context of Markov random fields.

- Is it clear how this work differs from previous contributions?
  - Yes. The setting itself is different from existing ones.

- Is related work adequately cited?
  - Yes. I think the work can benefit from referencing additional works. See below.


**Quality**

- Is the submission technically sound?
  - Yes. Proofs seem sound although I have not checked in full detail.

- Are claims well supported (e.g., by theoretical analysis or experimental results)?
  - Yes. Proos are provided in the appendix. Empirical validation is provided for synthetic and real-world data.

- Are the methods used appropriate?
  - Yes. The theoretical analyses use tools from optimization, linear algebra and concentration inequalities.

- Is this a complete piece of work or work in progress?
  - Complete work.

- Are the authors careful and honest about evaluating both the strengths and weaknesses of their work?
  - Discussions on the limitations of the presented approach can be beneficial.


**Clarity**

- Is the submission clearly written?
  - Yes.

- Is it well organized?
  - Yes. I was able to follow the main contributions without issues.

- Does it adequately inform the reader?
  - Yes. Experimental details are provided in the appendix. The work is reproducible.


**Significance**

- Are the results important?
  - Yes. I think the topic is of interest to the graphical models community.

- Are others (researchers or practitioners) likely to use the ideas or build on them?
  - Probably yes. I think the method proposed is an interesting direction to the setting under study.

- Does the submission address a difficult task in a better way than previous work?
  - The work can be considered as a first set of results for multitask DAG learning.

- Does it advance the state of the art in a demonstrable way?
  - Yes.

# Questions / Comments

1. I believe the work might benefit from a discussion section on the limitations of the results. For instance, Condition 3.3 (Bounded spectrum) might be an assumption that will not work for the high-dimensional case. In such scenario, due to n << p, there will be zero eigenvalues and the condition will not hold. Can authors comment on this?

2. The statement of Theorem 3.2. has been updated in the appendix, where authors claim that the appendix version is better than the one in the main paper. I am not sure if one is allowed to update this much a relevant part of the contributions, which is a major concern for me.

3. I think the paper can benefit from referencing works that also have a "direct" or "joint" goal instead of separate estimations. For instance:

    * In undirected networks: "Learning Shared Subgraphs in Ising Model Pairs" by Varici et al.
    * In linear SEMs: "Direct Learning with Guarantees of the Difference DAG Between Structural Equation Models" by Ghoshal et al.

  In both works above, there is also evidence of the advantages of a joint objective instead of performing separate estimations.

Typos:
* L51: p was not introduced.
* L245: Appendix F not G.

---
I thank the authors for their response. After reading other reviews and responses, I feel more positive about the paper and will increase my score to 7. I think that the paper could benefit by adding discussions on some points brought by the reviewers.


**Time Spent Reviewing:**

3

---

> ### Author Response · Authors · 2021-08-11
> **Response to Reviewer Masb**
>
> We thank the reviewer for the overall positive comments, constructive suggestions on paper refinement, and references to related works! We answer the raised questions below. Hope they can address your concerns.
>
> ---
>
> **Q1:** *I believe the work might benefit from a discussion section on the limitations of the results. For instance, Condition 3.3 (Bounded spectrum) might be an assumption that will not work for the high-dimensional case. In such scenario, due to n << p, there will be zero eigenvalues and the condition will not hold. Can authors comment on this?*
>
> First, we will address your concern about the bounded spectrum condition.
>
> - In fact, Condition 3.3 is imposed on the population covariance matrix $\Sigma^{(k)}$, **not** on the **empirical** covariance matrix $\widehat{\Sigma}^{(k)}$, so this condition does not depend on samples. Therefore, the n << p  scenario isn’t a concern.
>
> - However, we do need to bound the eigenvalues of a submatrix $ \widehat{\Sigma}^k_{SS}$ of the empirical covariance matrix in the proof. For example, in the last line on page 25, we apply existing concentration bound to show $||\widehat{\Sigma}_{SS}^k{}^{-1} ||_2\leq 2 /\Lambda_\text{min}$ with high probability. Here $S$ indicates some sparse support set with $|S|$<<$p$.
>
> Second, we agree with you that it is beneficial to discuss more challenging settings that the current analysis has not yet extended to, which will require a significant amount of additional work and could be conducted in the future.
>
> - One has been mentioned in Sec 7: “The current work applies to DAGs that have a certain similarity in sparsity pattern. It will be interesting to consider whether the joint estimation without the group-norm (and without the union support assumption) can also lead to a similar improvement in causal order recovery.”
>
> - The other challenge is how to get rid of the equal variance condition (Condition 3.1). This condition is required by the analysis but is less practical. **However, we would like to kindly refer you to our response to Reviewer x4HL, where we have described how our analysis can be readily extended to more practical cases**. For even more general cases that require a lot more space to state, we plan to mention it in a Discussion section as you suggested and leave it to future work.
>
> ---
>
> **Q2:** *The statement of Theorem 3.2. has been updated in the appendix, where authors claim that the appendix version is better than the one in the main paper.*
>
> Thank you for the careful reading which we appreciated! When we re-organized the appendix after the main paper submission we realized that some simple modifications to the derivation can lead to a better bound. Therefore we choose to present it in the Appendix. Since this is an improved bound, we think it should improve the quality of the paper.
>
> Furthermore, we believe our most important contribution still lies in the causal order recovery analysis in Theorem 3.1 because it is the most challenging component in the DAG recovery problem.
>
> ---
>
> **Q3:** *I think the paper can benefit from referencing works that also have a "direct" or "joint" goal instead of separate estimations. For instance…*
>
> Thank you for referring us to these related works! We have included some works on the joint estimation of “undirected” graphical models but missed the ones you mentioned. We will include them in the ‘Multi-task learning’ paragraph.
>
>
> ** Finally, thank you for pointing out the typos. We will correct them to improve the clarity.

---

### Decision · Program_Chairs · 2021-09-27

**Decision:**

Accept (Poster)

**Comment:**

This was a very borderline paper. After the author response and discussion, the one reviewer recommending rejection raised their score to a weak accept. Please make sure to read all of the reviewer feedback carefully and make suggested/promised changes for the camera ready. In particular, reviewer x4HL had some additional followup concerns that should be addressed.